# FGFR inhibition blocks NF-κB-dependent glucose metabolism and confers metabolic vulnerabilities in cholangiocarcinoma

Yuanli Zhen [1,2,3,4], Kai Liu[5], Lei Shi[1,2,3,4], Simran Shah[1], Qin Xu [1,2,3,4], Haley Ellis[1,2,3,4], Eranga R. Balasooriya [1,2,3,4], Johannes Kreuzer[1,3], Robert Morris[1], Albert S. Baldwin[6], Dejan Juric [1,3], Wilhelm Haas[1,3] & Nabeel Bardeesy [1,2,3,4] ✉

Genomic alterations that activate Fibroblast Growth Factor Receptor 2 (FGFR2) are common in intrahepatic cholangiocarcinoma (ICC) and confer sensitivity to FGFR inhibition. However, the depth and duration of response is often limited. Here, we conduct integrative transcriptomics, metabolomics, and phosphoproteomics analysis of patient-derived models to define pathways downstream of oncogenic FGFR2 signaling that fuel ICC growth and to uncover compensatory mechanisms associated with pathway inhibition. We find that FGFR2-mediated activation of Nuclear factor-κB (NF-κB) maintains a highly glycolytic phenotype. Conversely, FGFR inhibition blocks glucose uptake and glycolysis while inciting adaptive changes, including switching fuel source utilization favoring fatty acid oxidation and increasing mitochondrial fusion and autophagy. Accordingly, FGFR inhibitor efficacy is potentiated by combined mitochondrial targeting, an effect enhanced in xenograft models by intermittent fasting. Thus, we show that oncogenic FGFR2 signaling drives NF-κB-dependent glycolysis in ICC and that metabolic reprogramming in response to FGFR inhibition confers new targetable vulnerabilities.

Intrahepatic cholangiocarcinoma (ICC; cancer of the hepatic bile ducts) is the second most common liver malignancy and has been rising in incidence for several decades[1,2]. The prognosis for advanced cholangiocarcinoma is poor, with a median survival of only ~12 months despite standard combination chemotherapy and immunotherapy[3]. The identification of activating genomic alterations of FGFR2 in ~12–15% of cases has led to an important advance in precision medicine approaches for this subset of patients with ICC[4–10]. The most common alterations are fusions, involving the entire coding sequence of FGFR2—excluding the final exon encoding an autoinhibitory domain—with various partners[11,12]. Pan-FGFR kinase inhibitor (FGFRi) treatment (with pemigatinib, infigratinib, or futibatinib) improves outcomes of patients with FGFR2 + ICC, leading to the FDA approval of these agents in the second-line setting and beyond[13–15]. However, objective response rates are only 23–43%, and the median progression-free survival is ~6–9 months, in part due to acquired resistance, commonly driven by secondary mutations in the FGFR2 kinase domain[16–21]. Moreover, drug holds and dose reductions are often necessary due to on- and off-target toxicities. These include hyperphosphatemia caused by inhibition of FGFR1 in the renal tubule as well as skin, nail, and eye toxicities, which appear to stem, at least in part, from FGFR2 inhibition[13–15]. In addition, while preliminary clinical data show that selective FGFR2

¹Krantz Family Center for Cancer Research, Massachusetts General Hospital, Boston, MA, USA. ²Center for Regenerative Medicine, Massachusetts General Hospital, Boston, MA, USA. ³Dept. of Medicine, Harvard Medical School, Boston, MA, USA. ⁴The Cancer Program, Broad Institute, Cambridge, MA, USA. ⁵Center for Computational and Integrative Biology, Department of Molecular Biology, Massachusetts General Hospital, Harvard Medical School, Boston, MA, USA. ⁶Lineberger Comprehensive Cancer Center, University of North Carolina at Chapel Hill School of Medicine, Chapel Hill, USA. ✉e-mail: Bardeesy.Nabeel@mgh.harvard.edu

inhibition (with RLY-4008) has higher response rates, toxicities remain challenging, and disease progression inevitably occurs[21].

Elucidating the oncogenic pathways controlled by FGFR2 fusions is needed to help inform advances in therapy. The normal physiological functions of FGFR2 are carried out by multiple effectors, including the MEK/ERK, PI3K, JAK/STAT, and PLCγ pathways[22]. Studies in patient-derived models of FGFR2-fusion+ ICC show that FGFR inhibition selectively suppresses SHP2/MEK/ERK signaling[16]. This effect is potentiated by co-inhibition of the EGFR/ERBB pathway, preventing feedback activation of MEK/ERK and increasing treatment efficacy[23]. However, the key downstream molecular targets and cell biological processes mediating the oncogenic activity of FGFR2−MEK−ERK signaling in FGFR2 + ICC remain to be defined. Previous research has shown that in both healthy and tumor tissues, FGFR modulates diverse transcriptional regulators to control proliferation and differentiation and that a central downstream function of the pathway involves reprogramming cell metabolism[24,25]. The transcriptional mediators and metabolic effects of FGFR signaling vary depending on the specific genetic context and tissue of origin. In brain tumors with FGFR3−TACC1 fusions, this pathway induces mitochondrial biogenesis and increases respiration via stimulation of the PGC1α transcription factor[26]. FGFR4 signaling in hepatocytes upregulates glycogen and fatty acid metabolism and suppresses bile acid metabolism via SREBP and LXR transcription factors[27]. In endothelial cells and FGFR-amplified lung cancer cells, the pathway activates aerobic glycolysis to support differentiation and cell growth[24,25]. Notably, inhibition of oncogenic pathways can trigger a range of metabolic adaptations as a form of feedback response, analogous to signaling feedback mechanisms. These metabolic changes—including shifts in nutrient acquisition and utilization and alterations in mitochondrial function—can restore homeostasis and compensate for oncogene inactivation, supporting cell survival[28]. Thus, deciphering the metabolic program controlled by FGFR2 fusions and characterizing the compensatory metabolic processes resulting from FGFR inhibition could guide the development of effective treatment combinations.

Here, we explore the transcriptional and metabolic consequences of FGFR inhibition across a set of patient-derived models of FGFR2 + ICC. We demonstrate that FGFR2-mediated activation of the NF-κB transcription factor is essential for driving high levels of glucose metabolism in these models. The shutdown of this pathway upon FGFR inhibition confers new metabolic vulnerabilities that can be further exploited by nutrient deprivation in vitro and dietary restriction in vivo. These analyses illuminate the oncogenic circuitry of FGFR2-fusion+ ICC and expose therapeutic opportunities emerging from the metabolic rewiring resulting from FGFR inhibition.

## Results

### FGFR inhibition represses glycolytic gene expression in FGFR2-fusion+ ICC

We utilized a series of patient-derived models to define the transcriptional response to inhibition of FGFR signaling in FGFR2-fusion+ ICC, both in vivo and in vitro (Fig. 1a and Table 1, see Methods for information on the FGFR inhibitors used). The in vivo studies employed two patient-derived xenograft (PDX) models (MG212 and MG69) that are sensitive to FGFR inhibition, with anti-proliferative effects and shutdown of downstream ERK signaling (Supplementary Fig. 1a–g). The in vitro studies used a set of patient-derived cell lines, including an FGFRi sensitive model (ICC13-7) (Supplementary Fig. 1h) and two models that show adaptive feedback activation of EGFR/ERBB signaling upon FGFR inhibition but are sensitive to dual FGFR + EGFR/ ERBB inhibition (ICC10-6 and ICC21)[23] (Supplementary Fig. 1i, j). RNA sequencing and Gene Set Enrichment Analysis (GSEA) revealed that FGFRi treatment prominently downregulated a glycolysis gene signature in both PDX models (using the Hallmarks database; Fig. 1b, c). Likewise, the glycolysis signature was strongly repressed after 24 h

FGFRi treatment of the ICC13-7 cell line in vitro (Fig. 1d). In this regard, hexokinase 2 (HK2), encoding the first rate-limiting enzyme in glycolysis showed strong suppression across models, and other genes involved in glycolysis were coordinately downregulated, whereas TCA cycle genes were not changed (Fig. 1e, f). HK2 mRNA decreases were evident as early as 4 h after FGFRi treatment in ICC13-7 cells, as well as in the partially resistant ICC21 and resistant ICC10-6 models, where the suppression of HK2 was potentiated by dual FGFR/ERBB inhibition (Supplementary Fig. 2a, b). HK2 was also prominently decreased at the protein level across models after 24 h treatment (Fig. 1g–i and Supplementary Fig. 2c). Thus, concordant in vitro and in vivo data show that FGFR inhibition (±ERBB inhibition) potently suppresses glycolytic gene expression across FGFR2-fusion+ ICC models.

### Sustained FGFR2-signaling maintains hyperactive glucose metabolism

Based on the transcriptomics data, we next explored the functional impact of FGFR inhibition on cell metabolism of the FGFR2-fusion+ ICC13-7 model by conducting targeted liquid chromatography– tandem mass spectrometry (LC–MS/MS) for steady-state levels of 106 metabolites (Fig. 2a). Examination of ICC13-7 cells after 24 h FGFRi treatment revealed broad reduction of many metabolites, with most significant effects on glucose metabolism intermediates (Fig. 2b). Moreover, enrichment analysis using the MetaboAnalyst platform identified glycolysis as the most altered pathway (Fig. 2c), with significant drops in Hexose-phosphate, Fructose-1,6-bisphosphate (FBP), Glyceraldehyde 3-phosphate (G3P)/dihydroxyacetone phosphate (DHAP), 3-Phosphoglyceric acid (3PG), and Phosphoenolpyruvic acid (PEP) (Fig. 2d). There were also reductions in hexosamine biosynthesis pathway (HBP), pentose phosphate pathway (PPP), and Tricarboxylic acid (TCA) cycle metabolites, which are fed by intermediates from glycolysis. Select nucleotides were also depleted, namely dTTP and dATP (Fig. 2d), and mild drops were seen for many amino acids (Supplementary Fig. 3a). Similar profiles were observed in ICC21 cells upon dual FGFR/ERBB targeting, with the most pronounced effects on glycolytic intermediates (Supplementary Fig. 3b, c). Collectively, these results reveal that FGFR kinase activity is required to drive high levels of glucose metabolism in FGFR2-fusion+ ICC.

The marked decreases in the products of upper glycolysis (Hexose-phosphate, FBP) and associated biosynthetic pathway (HBP, PPP) are consistent with the reduced expression of HK2 resulting from FGFR inhibition. In line with these changes, bioluminescence assay (Glucose Uptake-Glo) demonstrated a reduction in glucose uptake in both ICC13-7 and ICC21 cells (Fig. 2e and Supplementary Fig. 3d). Additionally, the Lactate-Glo assay showed decreased lactate levels in the media, indicative of a decrease in glycolysis (Fig. 2f and Supplementary Fig. 3e). We utilized the Seahorse metabolic analyzer to assess rates of glycolysis and respiration. FGFR inhibition significantly suppressed glycolysis as reflected by measurement of the extracellular acidification rate (ECAR) (Fig. 2g and Supplementary Fig. 3f). By contrast, both the basal and maximal oxygen consumption rate (OCR) was sustained (Fig. 2h). Adaptive changes that either maintain or boost OCR in response to oncogene inactivation have been described as an important feedback mechanism that sustains cell survival. Thus, whereas inhibition of FGFR2 fusions strongly suppresses glucose metabolism, compensating mitochondrial functions could protect cell viability.

We subsequently used uniformly labeled $^{13}C_6$-glucose (U-$^{13}C_6$-glucose) to trace glucose flux into different pathways in response to FGFR inhibition in ICC13-7 cells (Fig. 2i, j). Cells were treated with FGFRi or vehicle for 24 h followed by 1 h or 24 h of labeling under the same treatment conditions. Analysis after 1 h demonstrated a pronounced reduction in utilization of U-$^{13}C_6$-glucose to produce FBP and GlcNAC-1P—with the former also decreased at 24 h (Fig. 2k and Supplementary Fig. 3g–i), indicating that FGFR signaling is critical to sustain glycolytic and HBP activity. Moreover, flux of glucose into the TCA cycle was also

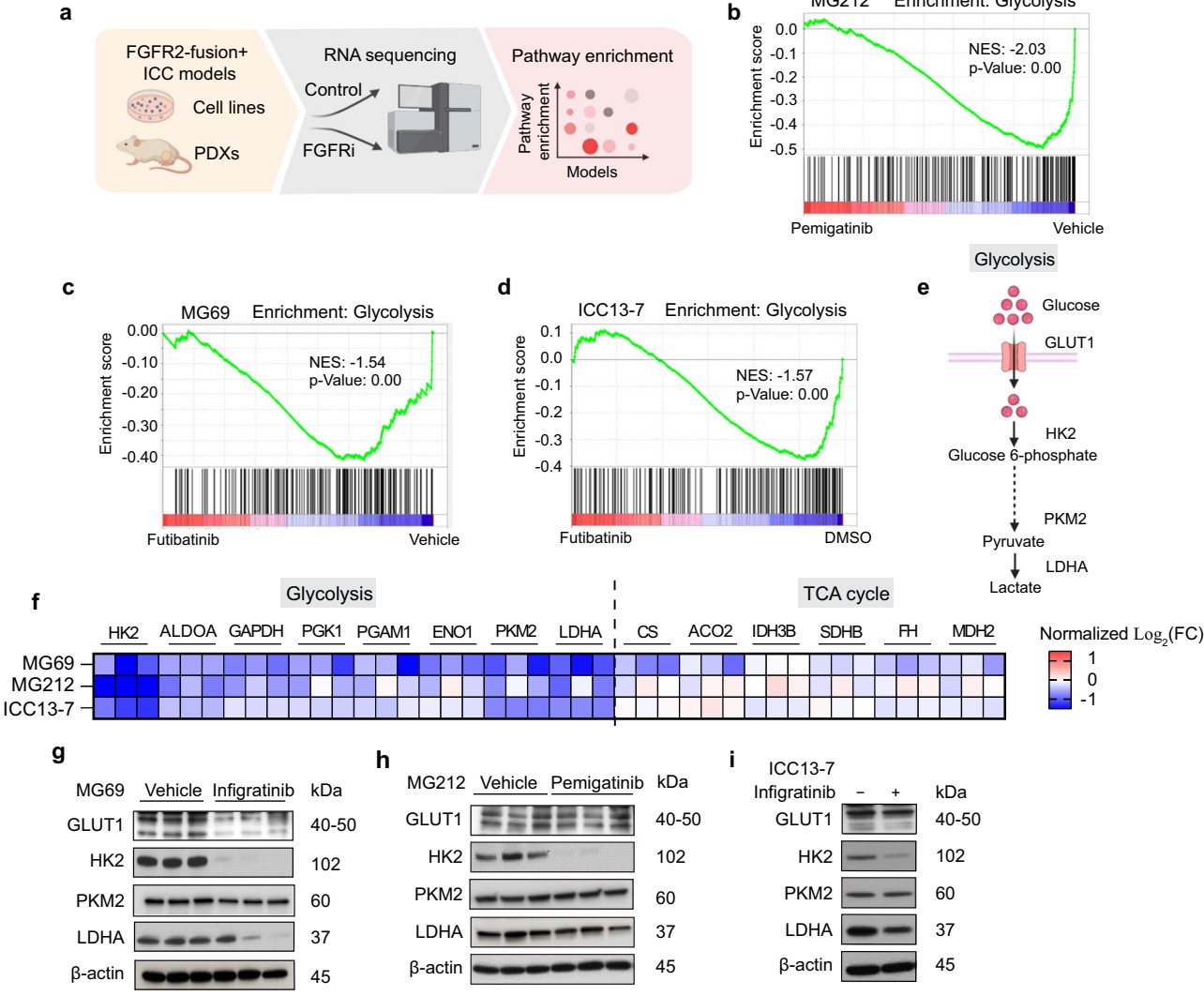

**Fig. 1 | FGFR inhibition represses glycolytic gene expression in FGFR2-fusion + ICC.** **a** Schematic of workflow. **b**–**d** GSEA (gene set enrichment analysis) of RNA-sequencing profiles using the Hallmark database, showing glycolysis pathway downregulation in the MG212 PDX model treated with pemigatinib (1 mg per kg, daily) versus vehicle for 11 days (**b**), in the MG69 PDX model treated with futibatinib (25 mg per kg, daily) versus vehicle for 14 days (**c**), and in ICC13-7 cells treated with 75 nM futibatinib versus DMSO for 24 h (**d**). The vertical black lines indicate the position of each of the genes of the studied gene set, and the green curve corresponds to the enrichment score (ES) curve obtained from GSEA software. NES normalized enrichment score. **e** Schematic of glycolysis pathway. **f** Heatmap of

changes in mRNA expression of enzymes in glycolysis and TCA cycle upon FGFR2 inhibitor treatment in the indicated models. Data are normalized to the control condition and presented as $Log_2$ transformation in each model. TCA: tricarboxylic acid; FC: fold change. **g**–**i** Immunoblot of the indicated proteins in the MG69 PDX treated with infigratinib (15 mg per kg, daily) or vehicle for 10 days (**g**), MG212 PDX treated with pemigatinib or vehicle for 11 days (**h**), and ICC13-7 cells treated with 100 nM infigratinib or DMSO for 24 h (**i**). For the mice experiment, $n = 3$ per group. A representative example of three replicates is shown for (**i**). **a**, **e** Created with BioRender.com. Western blots were repeated three times. Source data are provided as a Source Data file.

decreased, with significant decreases in M + 2 labeled citrate/isocitrate, malate and α-KG (Fig. 2k and Supplementary Fig. 3j). PPP activity appeared unchanged at the time points analyzed (Supplementary Fig. 3k). Thus, the FGFRi-driven decreases in steady-state levels of metabolites from these pathways are linked with a reduced rate of

production from glucose. Notably, large-scale mapping of cancer dependencies showed that FGFR2-fusion+ ICC cells have significantly enriched dependency on SLC2A1 (encoding GLUT1, the high-affinity glucose transporter) relative to other cancer cell lines (Fig. 2l; see Methods), consistent with a particularly high demand for glucose at baseline. Collectively, the data reveal that FGFR2 signaling stimulates glycolytic gene expression and reprograms glucose metabolism to support the growth of FGFR2-fusion+ ICC.

### FGFR2-mediated reprogramming of glucose metabolism requires NF-κB activation

We subsequently explored the mechanisms by which FGFR2 signaling promotes high levels of glucose metabolism in ICC cells. First, consistent with the MEK/ERK pathway being the primary effector of FGFR2 fusions, treatment with the SHP2 inhibitor, GDC-1971, or the MEK inhibitor, trametinib, decreased HK2 expression and potently

**Table 1 | Summary of FGFR2-fusion+ ICC models in the study**

| Model name | | FGFR2 fusion | Sensitivity to FGFRi |
|---|---|---|---|
| Cell lines | ICC13-7 | FGFR2-OPTN | Yes |
| | ICC21 | FGFR2-CBX5 | Partial, yes to FGFRi+ERBBi |
| | ICC10-6 | FGFR2-PHGDH | No, yes to FGFRi+ERBBi |
| PDX | MG212 | FGFR2-SORBS1 | Yes |
| | MG69 | FGFR2-KIAA1217 | Yes |

*PDX* patient-derived xenograft.

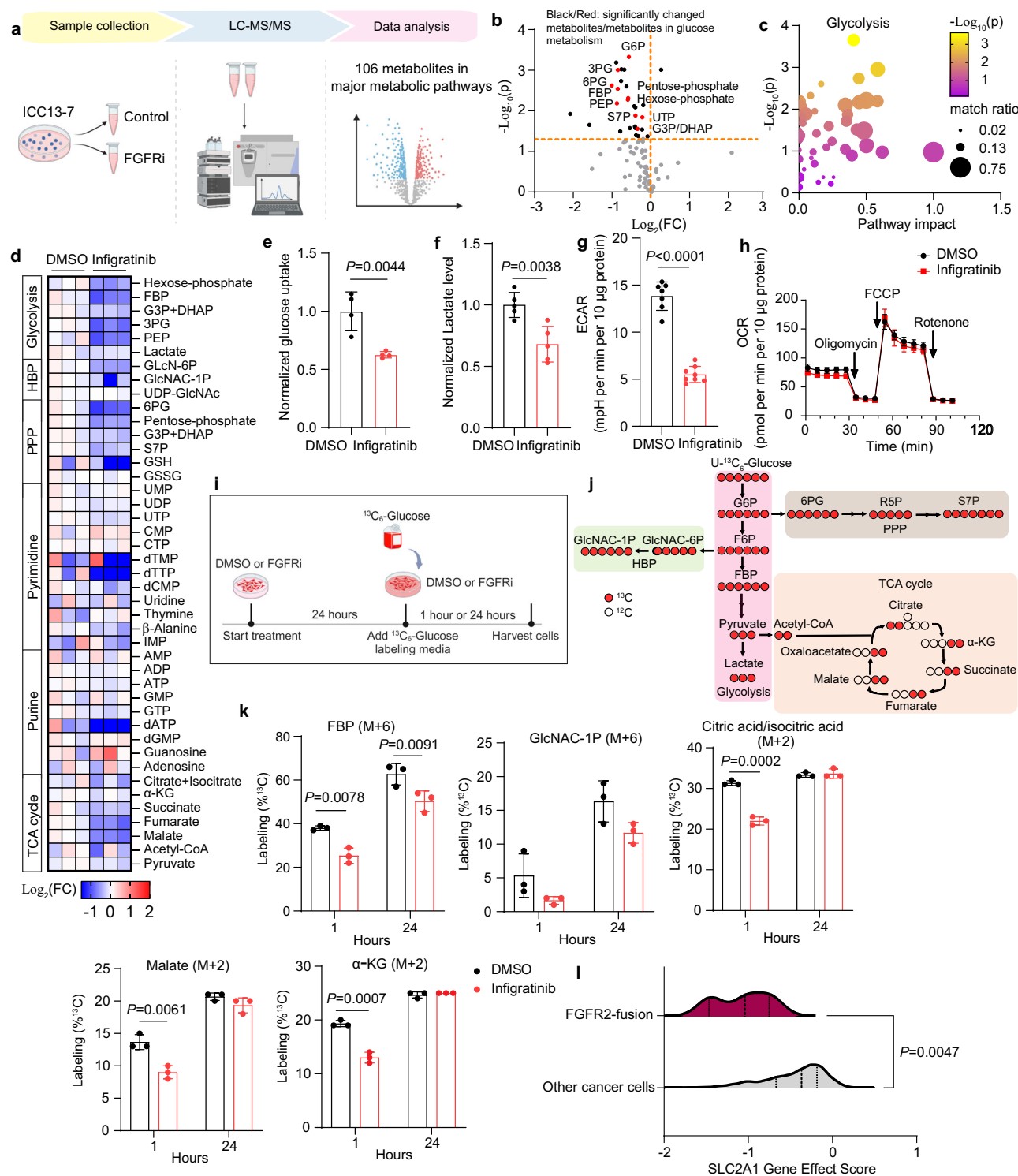

reduced glucose uptake and ECAR (Supplementary Fig. 4a–e). We next focused on the key mediators of the transcriptional program downstream of FGFR2-SHP2-MEK. Prediction of transcription factor activity changes upon FGFR inhibition ±ERBB inhibition, by interrogation of the differentially expressed genes using the TRRUST (transcriptional regulatory relationships unraveled by sentence-based text-mining) database, showed that treatment strongly downregulated NF-κB target gene signatures (i.e., RELA and NF-κB1 targets) across each of the models (Fig. 3a; representative targets are shown in Fig. 3b; RNAseq datasets are from Fig. 1 and published datasets[23]). Luciferase reporter assays in ICC13-7 cells confirmed that

NF-κB transcriptional activity was significantly reduced upon FGFRi treatment (Fig. 3c).

Subsequent western blot analysis showed that FGFR signaling strongly regulated the NF-κB pathway. In each cell line and PDX model, FGFRi treatment (or dual FGFR/ERBB inhibitor treatment) markedly reduced levels of activating phosphorylation of Inhibitor of κB kinase α/β (IKKα/β [phospho-S176/180]), which mediates NF-κB activation[29] (Fig. 3d, e and Supplementary Fig. 5a, b). The reduction in phospho-IKK was rapid, observed within 1 h of treatment, associated with progressive accumulation of the NF-κB inhibitory protein, IκBα, and gradually decreased levels of NF-κB inducing kinase (NIK) and of the NF-κB

**Fig. 2 | Sustained FGFR2-signaling maintains hyperactive glucose metabolism.**
**a** Schematic of workflow. FGFRi FGFR inhibitor. **b–d** Metabolite levels were
determined by LC−MS (liquid chromatography−mass spectrometry) of ICC13-7
cells treated with 100 nM infigratinib or DMSO for 24 h. **b** Volcano plot indicating
the metabolite changes. Metabolites in glucose metabolism showing significant
changes are represented as red dots and are identified by name. Other significantly
changed metabolites are shown as black dots. FC fold change. **c** Analysis of enri-
ched metabolic pathways by the MetaboAnalyst platform. **d** Heatmap depicting
metabolite changes from different pathways. Data are normalized to the DMSO
condition and presented as Log$_2$ transformation. **e, f** Relative changes of glucose
uptake ($n = 4$ biological replicates) (**e**) and lactate level in the medium ($n = 5$ bio-
logical replicates) (**f**) in ICC13-7 cells treated with 100 nM infigratinib or DMSO for
24 h. **g, h** Relative changes in extracellular acid rate (ECAR) (**g**) and oxygen

consumption rate (OCR) (**h**) normalized to protein amount in ICC13-7 cells treated
with 100 nM infigratinib ($n = 8$ samples) or DMSO ($n = 7$ samples) for 24 h.
**i** Schematic of workflow for the $^{13}C_6$-glucose tracing experiment. **j** Tracer scheme
illustrating the flux of U-$^{13}C_6$-glucose to different glycolytic branches. Red circle: $^{13}$C;
hollow circle: $^{12}$C. HBP hexosamine biosynthesis pathway; PPP pentose phosphate
pathway, TCA tricarboxylic acid. **k** $^{13}$C enrichment of different metabolites after
U-$^{13}$C-glucose labeling for 1 or 24 h in ICC13-7 cells, which were treated with 100 nM
infigratinib or DMSO for 24 h prior to labeling ($n = 3$ biological replicates). **l** Analysis
of SLC2A1 dependency in FGFR2-fusion ICC (ICC13-7, ICC10, ICC10-6) and other
cancer cell lines from Broad Institute Dependency Map (DepMap). Scores less than
−0.5 denote essentiality. Data represent means ± SD. Student's t-test (two-tailed)
was performed. **a, i, j** Created with BioRender.com. Source data are provided as a
Source Data file.

subunit precursor, p100 (Fig. 3d). In the PDX models (MG212 and
MG69), phospho-RELA (S536) levels were also dampened upon FGFR
inhibitor treatment (Fig. 3e and Supplementary Fig. 5a). In addition,
RELA showed progressive cytoplasmic retention and decreased
nuclear localization, while total protein levels did not change (Sup-
plementary Fig. 5c, d). SHP2 inhibition (GDC-1971) or MEK inhibition
(trametinib) also reduced phospho-IKK levels (Supplementary Fig. 5e).
We also confirmed that trametinib suppressed NF-κB target gene
expression (Supplementary Fig. 5f).

Global quantitative mass spectrometry-based phosphopro-
teomics of FGFR2-fusion+ ICC cells further highlighted links between
FGFR2 signaling and regulation of the NF-κB pathway. Cells were
treated with vehicle or with futibatinib for 4 or 24 h and then analyzed.
These studies detected 19,136 phosphorylation sites from 3954 pro-
teins (Fig. 3f). We then used the Kinase Library database[30] to predict
the kinases with the highest likelihood of phosphorylating the differ-
entially regulated phosphoserine and phosphothreonine (pS/T) sites
(Supplementary Fig. 5g). Enrichment analysis confirmed that the
activity of the ERK-RSK-(mTOR)-S6K axis was strongly reduced by
FGFRi treatment (Supplementary Fig. 5h, i). Notably, pathway analysis
of proteins with differential pS/T levels (utilizing the EnrichR platform)
revealed a significant reduction in the phosphorylation of NF-κB
pathway components after 4 h of FGFRi treatment (Fig. 3g, h). Several
of these changes involved predicted ERK or S6K sites, namely in the
IKK inhibitor TNIP1/ABIN1 (S619; S627), TNF receptor superfamily
member 21 (TNFRSF21 [S525]), and NF-κB2 (S161) (Supplementary
Fig. 5j–m), whereas phosphorylation of TNRFS21 (S565) and NF-κB2
(S222), which are non-ERK/S6K-related sites, were not changed at
either time point (Supplementary Fig. 5n). Moreover, FGFR inhibition
did not markedly affect the mRNA levels of NF-κB pathway receptors
or ligands (other than TNFSF15, whose receptor, TNFRSF25 shows very
low mRNA expression in ICC cells) (Supplementary Fig. 5o). Collec-
tively the data suggest that the NF-κB pathway was primarily regulated
by direct FGFR2-MEK-ERK signaling mechanisms rather than control of
expression of pathway components.

The NF-κB pathway can contribute to oncogenesis via diverse
mechanisms, including altering cell metabolism[31]. With respect to
metabolism, NF-κB has context-specific functions, either opposing or
enhancing cellular reprogramming to aerobic glycolysis[32–34], prompt-
ing us to examine the pathway further. Notably RELA knockdown or
treatment with the IKKβ inhibitor, TPCA-1, downregulated HK2 and
LDHA mRNA and protein levels (Fig. 3i–l and Supplementary Fig. 6a).
Moreover, knockdown of RELA or NIK (MAP3K14), or treatment with
TPCA-1, reduced glucose uptake and glycolysis in ICC13-7 and
ICC21 cells (Fig. 3m−q and Supplementary Fig. 6b–d). Furthermore,
proliferation analysis showed that FGFR2 fusion+ ICC cells had
increased sensitivity to RELA ablation or TPCA-1 treatment compared
to a set of non-FGFR-driven ICC cell lines (Supplementary Fig. 6e, f).
Thus, FGFR2-MEK-mediated activation of NF-κB is required to maintain
hyperactivated glucose metabolism and support the growth of FGFR2
fusion+ ICC models.

## Inhibition of FGFR signaling leads to adaptive metabolic changes that maintain mitochondria respiration in FGFR2-fusion + ICC

We next considered whether the altered metabolic state elicited by
FGFR2-MEK-NF-κB signaling could be exploited therapeutically. Ana-
logous to the feedback signal transduction processes that can offset
the shutdown of oncogenic signaling pathways, adaptive metabolic
shifts can sustain cell viability upon loss of oncogene-driven metabolic
reprogramming. In this regard, our finding that glucose flux into the
TCA cycle was impaired in response to FGFR inhibition in FGFR2 + ICC
cells, whereas mitochondria oxygen consumption was not changed,
suggested that other mitochondrial fuels may compensate for the
decrease in glucose metabolism. To further characterize the change of
mitochondrial fuel usage upon FGFR inhibition in FGFR2-fusion+ ICC,
we performed the Agilent Seahorse XF Mito Fuel Flex test, which
measures the rate of oxidation of different fuels by assessing the effect
of specific metabolic inhibitors on mitochondrial respiration (Fig. 4a).
By blocking specific pathways required for utilization of different
mitochondrial fuels (pyruvate, glutamine, and fatty acid), this assay
provides a measure of the dependency and capacity of cells to oxidize
each mitochondrial fuel (defined as importance of a given pathway for
maintaining respiration [dependency] and ability of mitochondria to
oxidize fuel in the context of inhibition of other pathways [capacity];
see Methods). Results from this test indicated that FGFR inhibition in
ICC cells results in increased dependency on fatty acid utilization
(Fig. 4b and Supplementary Fig. 7a). In this regard, in contrast to the
pronounced reduction in glucose metabolism, the fatty acid oxidation
(FAO) rate was either increased (ICC21) or unchanged (ICC13-7) upon
FGFR inhibition (Fig. 4c and Supplementary Fig. 7b). Of note, at the
transcriptional level, genes encoding key rate-limiting enzymes in fatty
acid synthesis were downregulated (Fig. 4d), while FAO genes were
upregulated (Fig. 4e).

FAO is fueled by fatty acids derived from lipolysis of intracellular
stores (lipid droplets) and from fatty acids and lipoproteins taken up
by the cell. We examined lipid utilization further by evaluating changes
in lipid droplet breakdown. BODIPY staining and quantification by
confocal microscopy showed that FGFRi treatment resulted in the
progressive and striking decrease in lipid droplet number from 24 to
48 h as well as a more gradual reduction in droplet size (Fig. 4f).
Consistent with these changes reflecting increased lipolysis, we found
that lipase activity was elevated by FGFRi treatment (Fig. 4g).

Lipolysis is mediated by PNPLA2 (adipose triglyceride lipase
[ATGL], the rate-limiting enzyme), LIPE (hormone-sensitive ligase
[HSL]), and MGLL (monoacylglycerol lipase). We predicted that
examination of the phosphoproteome at early and later time points of
FGFR inhibition (4 h and 24 h) would provide information about both
acute signaling changes proximal to FGFR2/ERK activity as well as
adaptive signaling changes reflecting alterations in the cell state. This
analysis revealed that phosphorylation of PNPLA2 (Ser-404) and LIPE
(Ser-855 and Ser-895) increased after 24 h of FGFRi treatment (Fig. 4h).
Phosphorylation of the conserved site Ser-406 of the mouse PNPLA2

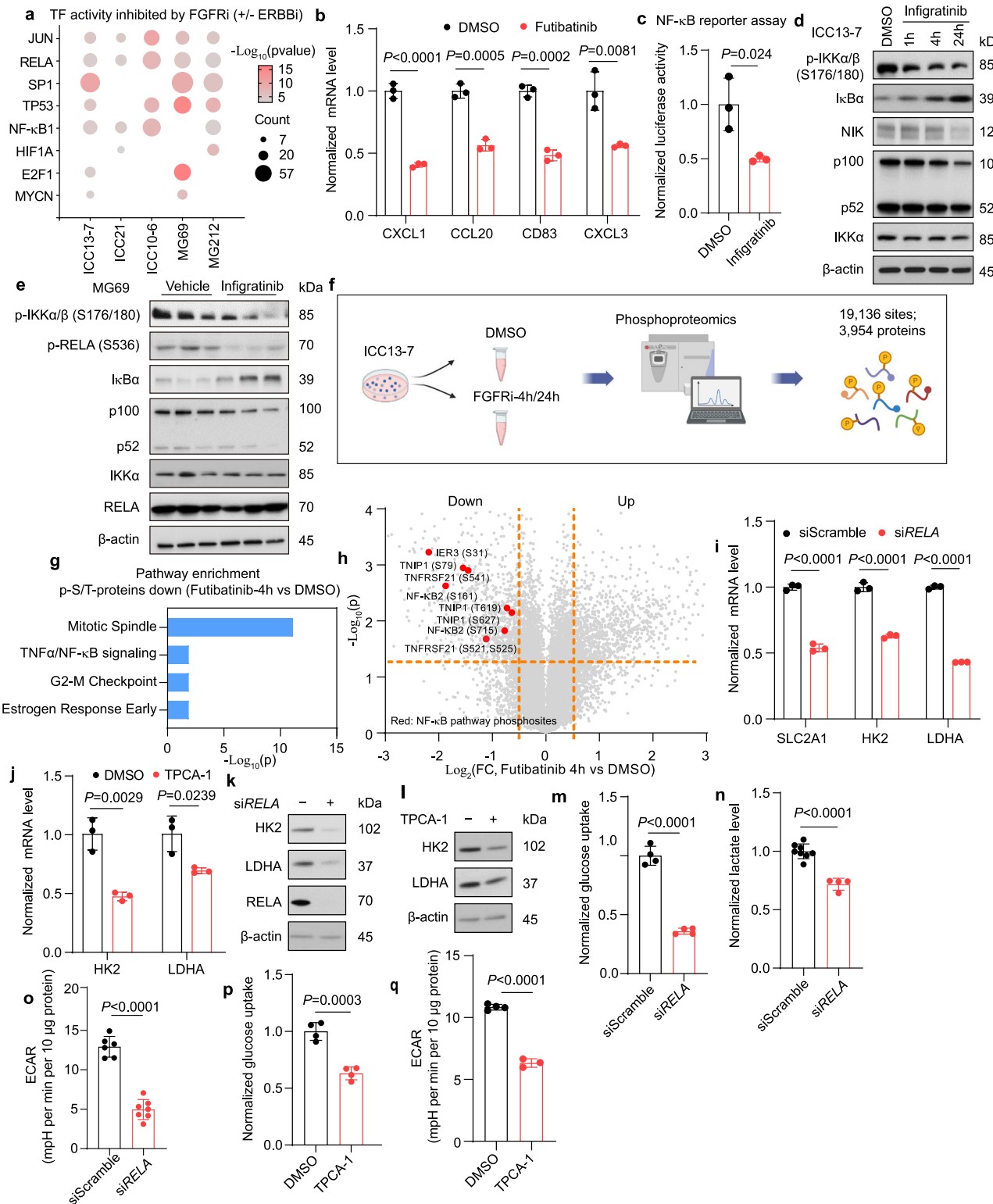

paralog by the energy-sensing kinase, AMPK, has been reported to stimulate lipolysis[35]. Like PNPLA2 (Ser-404), LIPE (Ser-855) is a predicted AMPK site (Supplementary Fig. 7d, e). Notably, PNPLA2 (Ser-404) and LIPE (Ser-855) phosphorylation were delayed, showing no changes after 4 h treatment, and furthermore, phosphorylation of the non-AMPK sites, PNPLA2 (Ser-428) and LIPE (Ser-951) was not significantly altered at either time point (Fig. 4i). Collectively, these data suggest that adaptive processes lead to a metabolic switch stimulating lipolysis in response to FGFR inhibition.

To monitor the utilization of exogenous fatty acids, we conducted U-$^{13}C_{16}$-palmitate tracing in FGFR2 + ICC cells ±FGFRi treatment. ICC13-7 cells were treated with DMSO or infigratinib for 24 h in lipid-depleted media and then switched to $^{13}$C-palmitate labeling media for another 8 h (with DMSO or FGFRi). Cells were then harvested for metabolomics (see schematic of experimental design in Supplementary Fig. 7f). FGFR inhibition did not affect the generation of palmitate-derived TCA cycle metabolites (Supplementary Fig. 7g). These data suggest that endogenous lipids, rather than exogenous lipids, may be the primary source

**Fig. 3 | FGFR2-mediated reprogramming of glucose metabolism requires NF-κB. a** Downregulated transcriptional regulators in ICC models upon FGFR (±ERBB) inhibition based on the TRRUST database. Treatments: ICC13-7: 75 nM futibatinib or DMSO (24 h); ICC10-6 and ICC21: DMSO, 100 nM infigratinib, 100 nM afatinib, or the combination (4 h); PDX-MG212: 1 mg per kg pemigatinib or vehicle (11 days); PDX-MG69: 25 mg per kg futibatinib or vehicle (14 days). **b** Relative mRNA levels of NF-κB targets in ICC13-7 cells treated with 75 nM futibatinib or DMSO for 4 h (n = 3 biological replicates). **c** Normalized NF-κB-dependent luciferase activity in ICC13-7 cells treated 24 h with vehicle or 100 nM infigratinib (n = 3 biological replicates). **d, e** Immunoblot of NF-κB pathway proteins in ICC13-7 cells treated with DMSO or 100 nM infigratinib for the indicated times (**d**), and in MG69 treated with vehicle or infigratinib for 10 days (**e**). **f–h** Phosphoproteomics on ICC13-7 cells treated with vehicle or FGFRi. **f** Schematic of study. Created with BioRender.com. **g** Pathway enrichment analysis (Hallmarks database) shows that futibatinib (75 nM, 4 h) downregulates Ser/Thr phosphorylation of NF-κB components in ICC13-7 cells. **h** Volcano plot of Ser/Thr phosphoproteome changes in ICC13-7 cells treated with

75 nM futibatinib for 4 h. Red dots: significantly downregulated phosphosites on NF-κB components. **i** Relative mRNA expression of the indicated glycolytic genes in ICC13-7 cells transfected with siRNA against RELA or control (n = 3 biological replicates) (**i**), and in ICC21 cells treated with 5 μM TPCA-1 or DMSO for 24 h (n = 3 biological replicates) (**j**). **k, l** Immunoblot of ICC13-7 cells treated with siRNA against RELA or control (**k**), or with 5 μM TPCA-1 or DMSO for 24 h (**l**). **m–o** Relative changes of glucose uptake (n = 4 biological replicates) (**m**), lactate secretion (n = 8 samples for siScramble; n = 4 samples for si*RELA*) (**n**), and ECAR (n = 6 samples for si-scramble; n = 7 samples for si*RELA*) (**o**) in ICC13-7 cells treated with siRNA against RELA or control. **p, q** Relative changes in glucose uptake (n = 4 biological replicates) (**p**) and ECAR (n = 4 samples for DMSO; n = 3 samples for TPCA-1) (**q**) in ICC13-7 cells treated with 5 μM TPCA-1 or DMSO for 24 h. For mouse experiments, n = 3 per group. Phosphoproteomics was performed in duplicate. Bar graphs: data represent means ± SD. Student's *t*-test (two-tailed) was performed. Western blots were repeated three times. Source data are provided as a Source Data file.

---

of fatty acids supplying FAO upon FGFRi treatment, although other lipid sources (e.g., lipoproteins) may also contribute.

Moreover, the expression of fatty acid uptake factors, including transport proteins (FATPs), binding proteins (FABPs) and receptors, was unaffected or slightly decreased by FGFRi treatment, with a modest upregulation of the transporter SLC27A3 and CPT1B (both expressed at very low levels [CPM < 1]) and ACSS1 (Acetyl-CoA synthetase 1, contributing to mitochondrial uptake) also exhibited a small increase in expression (Supplementary Fig. 7h).

As an additional adaptive change, we observed alterations in mitochondria morphology that aligned with increased efficiency of mitochondrial respiratory function upon FGFR inhibition[36,37]. In particular, Mitotracker staining demonstrated that FGFR inhibitor treatment resulted in a transition to a more fused (or tubulated) mitochondrial network (Fig. 4j, k). This was associated with reduced activating phosphorylation of the mitochondria fission mediator, p-DRP1 at Serine-616 (Fig. 4l), whereas protein levels of the fusion regulators, MFN1, MNF2, and OPA1, were not markedly changed (Supplementary Fig. 7i). Finally, we observed significant enrichment of the autophagy/lysosomal transcriptional program upon FGFR inhibition (or dual FGFR/EGFR inhibition) (Fig. 4m and Supplementary Fig. 7j, k), reminiscent of the adaptive upregulation of autophagy/lysosomal activity observed as a protective response to KRAS-MEK inhibition in pancreatic cancer cells[38–40]. In support of the functional effect of FGFR inhibition on autophagy regulation, we observed increased autophagic flux as detected by LC3-GFP-mCherry reporter assay, associated with increased LC3 staining and coordinated elevation in LC3B-II and decreased p62 levels assessed by immunoblot (Fig. 4n–p and Supplementary Fig. 7l, m). In summary, our data indicate that a series of adaptive changes in cell metabolism arise in response to inhibition of FGFR signaling in FGFR2 fusion+ ICC cells, with alterations in fuel source utilization and mitochondrial dynamics contributing to the maintenance of oxidative phosphorylation (OXPHOS) in the absence of glucose uptake.

**Targeting adaptive metabolic pathways increases the effectiveness of FGFR inhibitor treatment in FGFR2 fusion + ICC cells**

Informed by the metabolic plasticity observed upon FGFR inhibition in FGFR2 fusion+ ICC cells, we next explored the potential of targeting emerging metabolic vulnerabilities to improve treatment efficacy. To this end, we tested the effect of combined inhibition of FGFR and mitochondria oxidative metabolism. We first tested ONC212, an agonist of the ClpP protease, critical for mitochondrial protein quality control, which is currently in clinical development[41,42]. ONC212 potentiated FGFRi efficacy in ICC13-7 and ICC21 cells (Fig. 5a, top, and Supplementary Fig. 8a). Since glucose deprivation has been shown to sensitize cancer cells to OXPHOS inhibition, particularly in the setting of impaired glucose utilization[43], we also examined the impact of

reduced glucose conditions on these responses (11 mM glucose versus 1 mM glucose). We found that the cooperativity between ONC212 and FGFRi was augmented in low glucose media (Fig. 5a, bottom). Consistent with ClpP activation, ONC212 treatment led to a sharp decrease in levels of the ClpX chaperone protein and reduced mitochondrial oxygen consumption rate (Supplementary Fig. 8b, c).

We tested additional inhibitors of mitochondrial metabolism for cooperative effects. We found that FGFR inhibitor efficacy was also potentiated by combination treatment with the mitochondrial respiratory chain complex I inhibitor, IACS-010759[44] (Supplementary Fig. 8d). Moreover, based on the observed increase in metabolic dependency of FGFR2 + ICC cells on fatty acid oxidation (FAO) following FGFR inhibitor treatment (Fig. 4b), we also explored the cooperative impact of suppression of FAO via treatment with etomoxir, a CPT1A inhibitor. In both FGFR2 fusion+ ICC models tested (ICC13-7 and ICC21), combination treatment potentiated cell growth arrest compared to FGFR inhibitor alone (Supplementary Fig. 8e). Similar cooperative effects were seen in combination with chloroquine, an inhibitor of lysosomal acidification that blocks autophagy (Supplementary Fig. 8f). Thus, targeting adaptive changes in cell metabolism can augment the activity of FGFRi in FGFR2-fusion+ ICC cells, an effect that is enhanced by glucose deprivation.

**The therapeutic efficacy of FGFR inhibition against FGFR2-fusion + ICC xenografts is potentiated by mitochondrial targeting and fasting**

As in patients, FGFR inhibition in many xenograft models of FGFR2-fusion+ ICC results in modest shrinkage or stable disease rather than deep regressions, providing opportunities to investigate combination therapies[23]. Accordingly, we next examined FGFR2-driven metabolism and associated vulnerabilities in ICC xenografts. We found that FGFR inhibitor treatment (1 mg per kg pemigatinib) strongly attenuated glucose uptake in ICC13-7 tumors based on Flourine-18 fluorodeoxyglucose positron emission tomography/computed tomography ([18]F-FDG PET/CT), consistent with FGFR-dependent glucose uptake in vitro (Fig. 5b). We also conducted mass spectrometry-based metabolomics on this xenograft model after 7 days treatment with pemigatinib or vehicle. The data show a broad decrease in metabolites from cell carbon metabolism, with strong enrichment for glycolysis (Fig. 5c). Notable changes included drops in 6-PG, PEP, GlcNA-1-P, and FBP (Fig. 5d). These findings support a major role of the FGFR2-fusion in the metabolic reprogramming of ICC tumors.

To explore whether this FGFRi-driven impairment in glucose metabolism is exploitable, we tested the combined efficacy of FGFRi and mitochondrial targeting with ONC212 in ICC13-7 xenografts. Intermittent fasting has been shown to be a clinically applicable strategy for reducing glucose availability to potentiate OXPHOS inhibitor therapy. Therefore, based on these observations and our in vitro

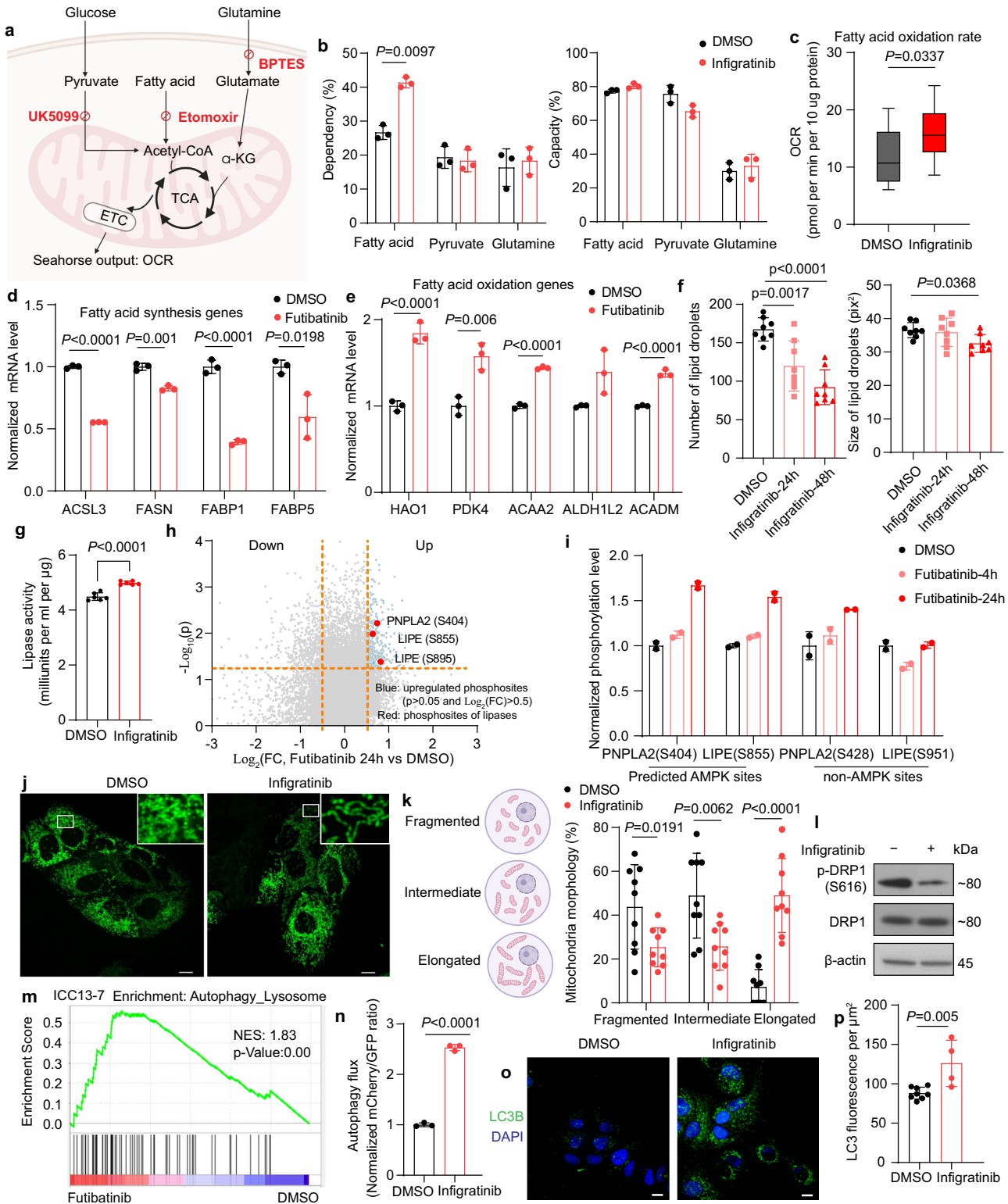

data showing improved efficacy of the FGFRi + ONC212 combination under reduced glucose conditions, we administered the treatments to mice that either received a regular diet or that were on an intermittent fasting regimen (Fig. 5e). This regimen reduced circulating glucose levels without causing significant weight loss (Fig. 5f, g). On both diets, single-agent FGFR inhibition (1 mg per kg pemigatinib, daily) blocked tumor growth but did not result in regressions, whereas ONC212 monotherapy had minimal effects (Fig. 5h, i and Supplementary Fig. 8g). By contrast, ONC212 potentiated FGFRi efficacy, leading to

tumor regressions, effects that were more marked with intermittent fasting compared to regular dietary conditions (Fig. 5h, i and Supplementary Fig. 8g, h). Consistently, combination treatment, when paired with intermittent fasting, induced a noteworthy reduction in tumor cell proliferation (Ki67 staining), surpassing the single agents (Fig. 5j, k) and outperforming the combination under standard dietary conditions (Supplementary Fig. 8i, j). Cleaved caspase-3 staining to assess cell death did not show significant differences between groups, although there was a trend suggesting an increase upon combination

**Fig. 4 | FGFR inhibition leads to adaptive metabolic changes that maintain mitochondria respiration in FGFR2-fusion + ICC. a** Schematic of Mito Fuel Flex Test. **b** Mitochondria fuel use capacity and dependency in ICC13-7 cells treated with 100 nM infigratinib or DMSO for 24 h ($n = 3$ biological replicates). **c** FAO rate in ICC21 cells treated with 100 nM infigratinib or DMSO for 24 h ($n = 12$ samples). The box plot shows the center line as the median, and the whiskers' boundary represents the minimum and maximum values of the dataset. The box extends from the 25th to 75th percentiles. **d, e** Relative mRNA expression of fatty acid synthesis (**d**) and FAO genes (**e**) in ICC13-7 cells treated with 75 nM futibatinib or DMSO for 24 h ($n = 3$ biological replicates). **f** Number and size of lipid droplets in ICC13-7 cells treated with DMSO or 100 nM infigratinib for 24 or 48 h, determined by BODIPY staining and confocal microscopy (n = 8 biological replicates). **g** Lipase activity in ICC13-7 cells treated with DMSO or 100 nM infigratinib for 48 h ($n = 6$ biological replicates). **h** Volcano plot indicating phosphoproteome changes in ICC13-7 cells treated with 75 nM futibatinib for 24 h. Upregulated phosphorylation sites ($\text{Log}_2(\text{FC}) > 0.5$ and $p < 0.05$) are shown in blue. Red dots: significantly upregulated phosphosites on lipases. **i** Phosphorylation changes of lipases after 4- and 24-h treatment with 75 nM futibatinib ($n = 2$ biological replicates). **j, k** Mitochondrial

morphology analysis (MitoTracker staining) of ICC13-7 cells treated with 100 nM infigratinib or DMSO for 24 h. **j** Representative images. Inset: higher magnification of boxed region. **k** Quantification of mitochondrial morphology states. Left, mitochondrial morphology scoring scheme. Right, Quantification ($n = 9$ images examined over 2 independent experiments). **l** Immunoblot of ICC13-7 cells treated with 100 nM infigratinib or DMSO for 4 h. Experiment repeated three times. **m** GSEA plots of Autophagy_Lysosome gene signature[69] in ICC13-7 cells treated with 75 nM futibatinib or DMSO for 24 h. **n** Autophagy flux analyses via flow cytometry using LC3-GFP-mCherry reporter in ICC13-7 cells treated with DMSO or 100 nM infigratinib for 48 h ($n = 3$ biological replicates). **o, p** Immunofluorescence staining of LC3B in ICC13-7 cells treated with DMSO or 100 nM infigratinib for 48 h. **o** Representative images. **p** Quantification of LC3B fluorescence intensity per $\mu\text{m}^2$ ($n = 8$ images for DMSO, $n = 4$ images for infigratinib over 2 independent experiments). Data represent means ± SD. Student's $t$-tests (two-tailed) were performed for (**b–e, g, h, k, n, p**). One-way ANOVA multiple comparisons were performed for (**f**). Scale bar: 10 μm. **a, k** Created with BioRender.com. Source data are provided as a Source Data file.

---

treatment with fasting (Supplementary Fig. 8k). We extended these studies to a second xenograft model, using patient-derived ICC21 cells. In the context of intermittent fasting, we found that whereas pemigatinib slowed growth and ONC212 had no effect, the combination potentiated tumor growth arrest (Supplementary Fig. 8l, m). Thus, we observed the efficacy of combined mitochondrial targeting and FGFR inhibition in xenografts generated from two independent patient-derived models of FGFR2-fusion+ ICC. Finally, we observed similar potentiation of tumor shrinkage in ICC13-7 xenografts by combination treatment with FGFRi and the complex I inhibitor, IACS-010759, on an intermittent fasting regimen (Supplementary Fig. 8n, o). Thus, FGFR2 inhibition potently attenuates glucose metabolism in vivo, sensitizing FGFR2 + ICC xenografts to a dual reduction in glucose availability and in OXPHOS activity.

## Discussion

The FDA approval of FGFR tyrosine kinase inhibitors (TKIs) has been a significant advance in the treatment of FGFR2 + ICC. Moreover, next-generation FGFR TKIs, which are more selective for FGFR2 and have an improved ability to overcome resistance mutations, are leading to higher response rates. However, even with these advances, acquired resistance to FGFR inhibitor monotherapy continues to develop within one year in most patients. It is clear that combination strategies will be necessary to achieve more durable efficacy. In this regard, there has been limited insight into the downstream molecular and cellular program driven by FGFR2 that could inform future therapeutic approaches. Here, we demonstrate that FGFR2 is critical for supporting high levels of glucose metabolism in FGFR2 + ICC cells, which requires NF-κB-driven glycolytic gene expression. Inhibiting FGFR down-regulates glycolysis, exposing a metabolic vulnerability. Combining mitochondrial-targeted treatments further enhances efficacy, especially when integrated with an intermittent fasting regimen (Supplementary Fig. 9). This work highlights a major function of oncogenic FGFR2 in metabolic reprogramming of ICC and suggests opportunities to harness the pronounced impairment of glucose utilization resulting from FGFR TKI treatment.

Although metabolic reprogramming is a central feature of cancer, often directly governed by driver oncogenes, there has been limited progress in developing new therapies that target cancer metabolism[45]. The significant plasticity of metabolic pathways can contribute to this challenge. In the case of FGFR inhibition, our work shows that while treatment attenuates glucose metabolism, compensatory processes sustain OXPHOS and support cell viability. The observed changes include increased reliance on fatty acids as a fuel source as well as induction of mitochondrial fusion and autophagy—both of which have been linked to increased respiratory efficiency in glucose-limited

conditions and other stresses[36,46,47]. Importantly, whereas OXPHOS inhibition is normally offset by adaptive increases in glycolysis, this flexibility is lost upon FGFR inhibition, leading to cooperative effects of combination treatment.

This emergent metabolic vulnerability prompted us to explore the additional benefit of dietary restriction. Epidemiological studies and experiments in model systems have shown that overall caloric intake and dietary composition can have a significant effect on cancer risk. Moreover, current investigations have emphasized the influence of diet on the growth of fully formed cancers and on treatment response[48–51]. In this regard, various modalities of dietary restriction can improve cancer therapies due to systemic hormonal and metabolic changes that influence growth factor signaling and nutrient availability in tumor cells. Intermittent fasting has been shown to provide a strategy to harness treatments that impair OXPHOS, with the reduction in glucose availability serving to constrain the adaptive response[52]. Both our in vitro and in vivo studies supported the beneficial effects of restricted glucose availability on the efficacy of treatments that jointly attenuate glycolysis (FGFRi) and OXPHOS (ONC212). Rationally designed metabolic therapies, combining dietary interventions with complementary systemic therapy, are an emerging treatment paradigm that could leverage nutrient limitation within the tumor and offer alternative approaches to exploit these metabolic shifts.

Our study reveals that the NF-κB pathway is an important effector of FGFR2 fusions in ICC, with FGFR inhibition leading to rapid loss of MEK-ERK signaling and consequent reduction in NF-κB activity. These findings contrast with several studies where receptor tyrosine kinases are connected to NF-κB through phospholipase C γ (PLCγ), which we have shown to not be controlled by FGFR2 fusions, consistent with deletion of the PLCγ binding site caused by the translocation. NF-κB is a central regulator of inflammation and contributes to the development and progression of many cancers by inducing immunomodulatory cytokines and altering cellular energy metabolism. In this regard, we found that NF-κB is required to support glycolysis in FGFR2-fusion+ ICC cells and to maintain expression of HK2. HK2 catalyzes the first committed step in glucose metabolism, phosphorylating glucose to produce glucose-6-phosphate (G6P), promoting further glucose uptake and providing the substrate for subsequent glucose utilization. We also find that the FGFR2-NF-κB signaling supports the expression of key inflammatory cytokines gene. Thus, this pathway may have joint roles in metabolic reprogramming and inflammation/immunomodulation in ICC. The key direct mediators of NF-κB regulation downstream of FGFR2, including the role of cytokine receptors, will require further study. Moreover, ERK phosphorylates diverse transcription factors, some of which are likely to also contribute to metabolic reprogramming in FGFR2-driven ICC.

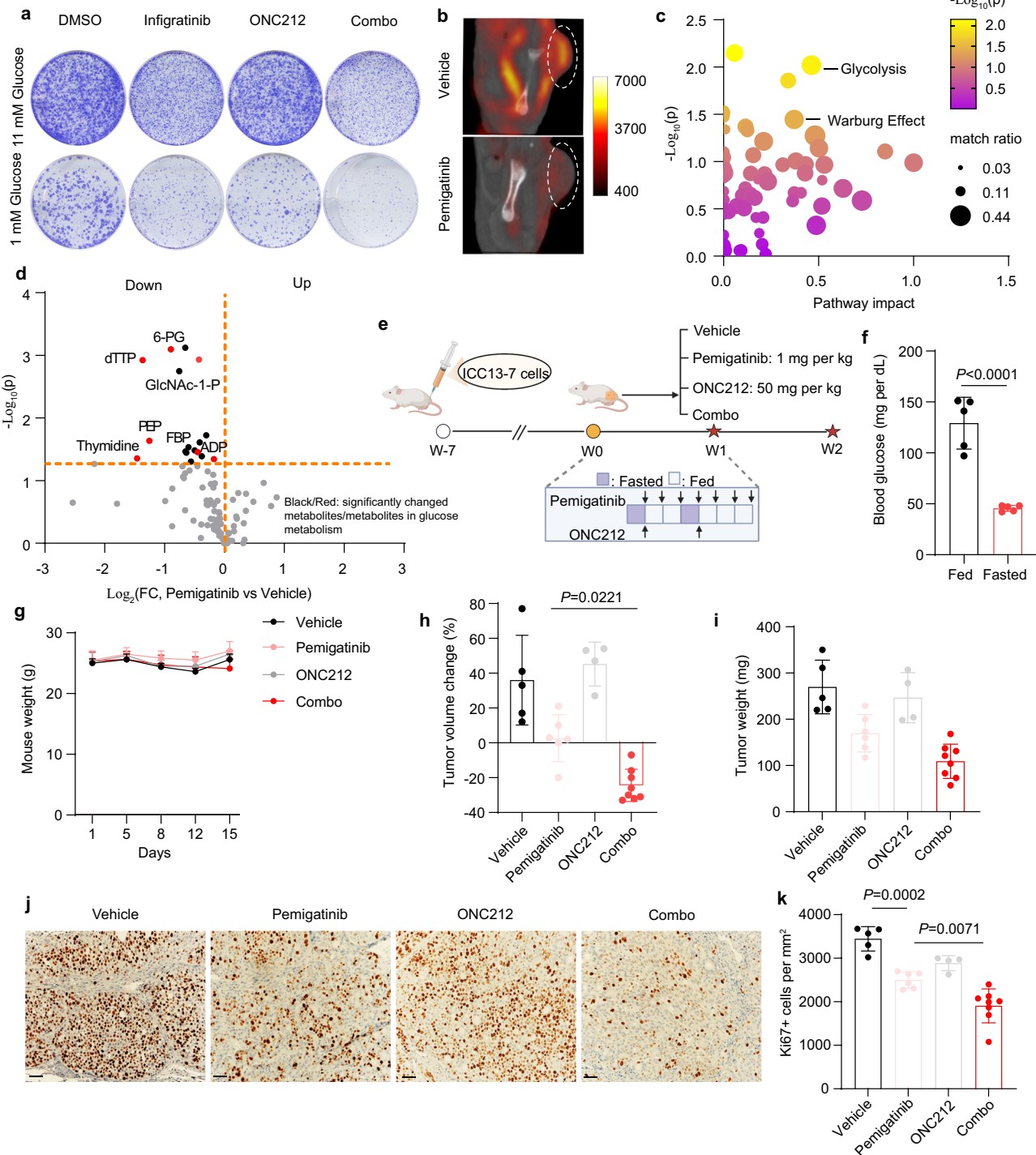

**Fig. 5 | Targeting adaptive metabolic pathways and glucose restriction increases the effectiveness of FGFR inhibitor treatment in FGFR2 fusion + ICC.** **a** Crystal violet staining to assess viability of ICC13-7 cells treated with DMSO, single agent infigratinib 50 nM, ONC212 20 nM, or the combination in high glucose (11 mM) and low glucose (1 mM) conditions. The data shown are representative of results from three independent experiments. **b** $^{18}$F-FDG PET/CT ($^{18}$F-fluorodeoxyglucose positron emission tomography and computed tomography) imaging of the ICC13-7 xenograft-bearing NSG mice upon 1 mg per kg pemigatinib or vehicle treatment for 4 days. Dashed lines highlight the tumors from xenografts. **c, d** Metabolite levels were determined by LC–MS of ICC13-7 xenografts from mice treated with pemigatinib 1 mg per kg ($n = 3$) or vehicle ($n = 3$) for 7 days. **c** Analysis of enriched metabolic pathways by the MetaboAnalyst platform. **d** Volcano plot indicating the metabolite changes. Metabolites in glucose metabolism showing significant changes are represented as red dots and are identified by name. **e** Schematic of intermittent fasting and treatment regimen. Created with BioRender.com. **f** Blood glucose concentration in mice harboring ICC13-7 xenografts. Mice were fed ad libitum or were fasted for 24 h ($n = 5$ mice per group). **g**–**i** Mice harboring ICC13-7 xenografts with a starting tumor volume ~200 mm$^3$ were treated daily with vehicle ($n = 5$), pemigatinib 1 mg per kg ($n = 6$), ONC212 50 mg per kg (n = 4), or the combination of both drugs (n = 8) for 15 days under the intermittent fasting regimen. Mouse weight (**g**), tumor volume change (**h**), and tumor weight (**i**) are shown. **j, k** IHC staining for Ki67 in ICC13-7 xenografts from the indicated treatment groups. **j** Representative images. **k** Quantification. Vehicle ($n = 5$), pemigatinib ($n = 6$), ONC212 ($n = 4$), and combo ($n = 8$). For bar graphs, data represent means ± SD. One-way ANOVA multiple comparisons were performed except for panel (**f**), which was calculated using the student's $t$-test (two-tailed). Scale bar: 50 μm. Source data are provided as a Source Data file.

We showed that FGFR inhibition led to a pronounced reduction in glucose uptake into FGFR2-driven ICC cells in vitro and loss of FDG-PET signal in xenograft tumors. Although PET/CT scans are rarely used for monitoring ICC treatment, the essential role of the FGFR2/MEK/ERK pathway in glucose uptake in FGFR2 + ICC may help in assessing therapeutic response. In a recent study analysis correlating pathological findings with pre-operative PET/CT scans, Vigano et al. describe an ICC patient with heterogeneous FDG uptake in their liver tumors[53]. Genomic analysis revealed the patient's tumor was heterogenous—the highly FDG avid liver lesion harbored an *FGFR2* translocation, whereas the hepatic lesion with low-level FDG avidity did not. In this regard, clinical resistance to FGFR TKI treatment is typically polyclonal and involves heterogeneous mechanisms, including secondary FGFR2 kinase domain mutations that prevent drug binding, genomic activation of 'bypass pathways' (e.g., KRAS mutations), and adaptive signaling mechanisms that sustain MEK/ERK pathway activity. FDG-PET may provide early information regarding the impact on different tumor nodules of systemic treatments designed to overcome resistance. PET/CT may be particularly helpful in assessing response or resistance to FGFRi in bone metastases that are common in advanced FGFR2+ cholangiocarcinoma[54] but cannot be easily evaluated by CT scans or MRIs.

In summary, we have identified FGFR2 fusions as potent drivers of aerobic glycolysis in ICC and revealed NF-κB as a key mediator of this process. Inhibition of FGFR leads to impaired glucose metabolism, limiting the adaptability of ICC cells and creating targetable metabolic vulnerabilities both in vitro and in vivo. These data support the further exploration of metabolic interventions, encompassing pharmacological and dietary strategies, to enhance FGFR inhibitor efficacy and potentially improve outcomes for patients with ICC.

## Methods

### Ethics statement

All research conducted for this manuscript complies with ethical regulations. All mouse experiments were conducted under protocols 2005N000148 and 2019N000116 approved by the Institutional Animal Care and Use Committee at Massachusetts General Hospital and comply with all regulations for the ethical conduct of research.

### FGFR inhibitors

These studies employed each of the three FGFR inhibitors that have been approved by the FDA for the treatment of FGFR2+ biliary tract cancers. All are highly selective and potent pan-FGFR inhibitors at the concentrations used[55]. These compounds were purchased from Selleckchem and include the reversible inhibitors, pemigatinib (S0088) and infigratinib (S2183), and the covalent inhibitor, futibatinib (S8848). These compounds have shared structural features, have low nanomolar potency against FGFR1–4 in cells (with pemigatinib and infigratinib, being moderately more selective for FGFR1–3), and show comparable biochemical and biological effects against cells harboring FGFR2 fusions lacking secondary kinase domain mutations[16,23]. The specific FGFRi employed in each experiment is indicated in the Figures and Legends.

### Cell culture

ICC13-7, ICC21, ICC10-6, ICC10, ICC10-8, and ICC12 are patient-derived ICC cell lines[23,56]. HUCCT1 was obtained from Riken Bioresource Center. CC-SW-1 and SG231 were from Dr. Theresa L. Whiteside (University of Pittsburgh). All the cell lines were cultured in RPMI (Corning, 10-041-cv) supplemented with 10% fetal bovine serum (Gibco, 26140-079), and 1% penicillin/streptomycin (Gibco, 15140-122), used within 10 passages and tested regularly for mycoplasma (Venor™GeM Mycoplasma Detection Kit, Sigma, MP0025). Cell counting was conducted with trypan blue exclusion on an automated cell counter (Invitrogen, Countess 3).

### Immunoblotting

Snap-frozen tumor tissues were homogenized using a Precellys 24 homogenizer (Bertin Instruments) and 2.8 mm disruptor beads (Bertin Instruments) with a setting of 6500 rpm 3 times in SDS lysis buffer (1% SDS, 10 mM HEPES [pH 7.0], 2 mM MgCl₂, 500 U Benzonase) with Halt™ Protease Inhibitor Cocktail (Thermo Fisher Scientific, 78,437) and phosphatase inhibitor cocktail set I and II (Millipore, 524624 and 524625). Cells were directly harvested in the lysis buffer after cold PBS washing. Total protein concentrations were measured with the BCA Assay Kit (Thermo Fisher Scientific, 23224). Then, equal amounts of protein were subjected to electrophoresis on 10% to 12% SDS–PAGE gels or on 4–20% Bio-Rad Mini-PROTEAN precast gels following the standard SDS-PAGE method, including transferring proteins to PVDF membranes (Millipore), blocking with a TBST buffer containing 5% skim milk, and incubating with primary antibodies dissolved in Universal Antibody Dilution Buffer (Millipore, U3635) overnight at 4 degree and secondary antibody for 1 h at room temperature. Finally, blots were developed using SuperSignal West Pico PLUS Chemiluminescent Substrate (Thermo Fisher Scientific, 34,580) and exposed on autoradiograph film (Denville Scientific, E3018) or the ChemiDoc image system (Biorad) according to the signal intensity.

### Antibodies and compounds

The primary antibodies used in the study included (The application is for Western blot unless specifically mentioned): from cell signaling technology, p-FRS2 (Y196) (3864), p-ERK1/2 (T202/Y204) (4370, clone D13.14.4E), ERK1/2 (4695, clone 137F5), p-RELA/p65 (S536) (3031), RELA/p65 (8242, clone D14E12), p-IKKα/β (S176/180) (2697, clone 16A6), IKKα (11930, clone 3G12), IkBα (4812, clone 44D4), NIK (4994), NF-kB2/p100/p52 (37359, clone D7A9K), β-Tubulin (86298, clone D3U1W), PARP (9542), HK2 (2867, clone C64G5), LDHA (3582, clone C4B5), PKM2 (4053, clone D78A4), p-DRP1 (S616) (3455), DRP1 (8570, clone D6C7), MFN1 (14739, clone D6E2S), MFN2 (9482, clone D2D10), OPA1 (80471, clone D6U6N), p62 (23214, clone D6M5X), LC3B (3868, clone D11) for immunoblot with dilution 1:1,000-1:3,000; from Millipore-Sigma, β-actin (A5316, clone AC-74) with dilution 1:20,000; from Abcam, FRS2 (ab183492), CLPX (ab168338, clone EP8772) with dilution 1:1,000; from Servicebio for immunofluorescence, KI67 (GB13030-2, 1:200); from Novus Biologicals for immunofluorescence, pan-cytokeratin (NBP2-29429, clone AE-1/AE-3, 1:200); from MyBioSource, GLUT1 (MBS9126610, 1:2,000); from MBL, LC3B (PM036, 1:200) for immunofluorescence. The secondary antibodies used in the study included: horseradish peroxidase (HRP)-conjugated secondary antibodies (Vector Laboratories: anti-rabbit, PI-1000; anti-mouse, PI-2000) with dilution 1:10,000. RELA/p65, NIK, and NF-κB2 (p100/p52) antibodies were validated in human samples for Western blot by gene knockdown. Validation of other commercial antibodies is available in the product page and search of relevant literature (details see Reporting Summary).

The compounds used in the study included: from MedChemExpress, TPCA-1 (HY-10074), Etomoxir (HY-50202), Chloroquine (HY-17589A), Trametinib (HY-10999); from Selleckchem, Afatinib (S1011); GDC-1971 (E1191); ONC212 was gifted by Chimerix, Inc.; IACS-10759 was gifted by Dr. Haoqiang Ying.

### Real-time qPCR and RNA sequencing

RNA was extracted from cells or tumor tissues under different treatment conditions with the RNeasy Plus Mini Kit (QIAGEN) in accordance with the manufacturer-prescribed protocol.

Total RNA was measured using a NanoDrop spectrophotometer (Thermo Fisher). Subsequently, 1 μg of RNA was used to generate cDNA via the SuperScript™ IV VILO™ Master Mix (Invitrogen, 11766050) as directed by the manufacturer. cDNA was amplified using Universal SYBR Green (Bio-Rad). Samples were processed using a CFX384 Touch Real-Time PCR Detection System (Bio-Rad). Relative mRNA expression

was normalized to Tubulin control. Primers used include: HK2 (Forward: GAGCCACCACTCACCCTACT; Reverse: CCAGGCATTCGG-CAATGTG), CXCL1 (Forward: AGTGGCACTGCTGCTCCT; Reverse: TG GATGTTCTTGGGGTGAAT), CD83 (Forward: TACAGAGCGGAGAT TGTCCTGC; Reverse: GCTCGTTCCATGCCAGCTTTAG), CCL20 (Forward: AAGTTGTCTGTGTGCGCAAATCC; Reverse: CCATTCCAGAAA AGCCACAGTTTT), CXCL3 (Forward: TTCACCTCAAGAACATCCAA AGTG; Reverse: TTCTTCCCATTCTTGAGTGTGGC), and Tubulin (Forward: ACCAACCTACGGGGATCTGAA; Reverse: TTGACTGCCA ACTTGCGGA).

RNA sequencing of ICC21 (treated with DMSO, Infigratinib, Afatinib, and Combo), ICC10-6 (treated with DMSO, Infigratinib, Afatinib, and Combo), MG212 (treated with vehicle and Pemigatinib) and ICC13-7 (transfected with siScramble and siRELA) was performed and analyzed at GENEWIZ (genewiz.com). RNA samples were quantified using a Qubit 2.0 Fluorometer (Life Technologies, Carlsbad, CA, USA). RNA integrity was determined using Agilent TapeStation 4200 (Agilent Technologies, Palo Alto, CA, USA). The NEBNext Ultra II RNA Library Prep Kit for Illumina was used based on the manufacturer's instructions (NEB, Ipswich, MA, USA). The sequencing libraries were clustered on flow cells and loaded onto an Illumina HiSeq (4000 or equivalent) according to the manufacturer's instructions. Sequencing was done using a 2 × 150 bp Paired End (PE) configuration. HiSeq Control Software (HCS) was used for image analysis and base calling. Raw sequence data (.bcl files) were converted into fastq files and de-multiplexed (Illumina's bcl2fastq 2.17). Trimmomatic v.0.36 was used to trim sequence reads to remove adapter sequences and nucleotides with poor quality. STAR aligner v.2.5.2b was used to map the trimmed reads to the Homo sapiens reference genome (ENSEMBL). DESeq2 was employed to compare expressions between the samples. $P$-values and Log$_2$ fold changes were generated using the Wald test.

RNA sequencing of ICC13-7 (treated with DMSO and Futibatinib) and MG69 (treated with vehicle and Futibatinib) was performed and analyzed at MGH next generation sequencing core. STAR v. 2.3.0 was used to map RNA-seq reads to the hg19 reference genome. HTSeq v. 0.6.0 was used to produce read counts for individual genes. The edgeR package v. 3.36.0 was used to perform differential expression analysis subsequent to read count normalization and restricted inclusion of genes with count per million reads (CPM) > 1 for at least one of the samples.

## Glucose uptake and lactate secretion measurement

In total, 10,000 cells per well were seeded in a 96-well plate and treated under the indicated conditions for 24 h. Cells were then subjected to the Glucose Uptake-Glo™ Assay (Promega, J1341) or Lactate-Glo™ Assay (Promega, J5021), performed according to the manufacturer's instructions.

## Targeted metabolomics

Sample preparation: a. ICC cells. 1 million ICC cells were seeded in 6 cm dishes under indicated treatments with triplicates to identify metabolic characteristics. Fresh media was provided 2 h before harvest. For metabolite collection, media was completely aspirated, and cells were washed with saline quickly. After washing, fully remove saline, and cells can be scrapped in 1 ml pre-cooled methanol (−20 °C) with internal standards (Cambridge Isotope Laboratories, MSK-A2-1.2), and transferred to glass vials, stored at -80 °C until extraction. b. ICC xenografts. Mice harboring ICC13-7 xenografts were treated with vehicle or pemigatinib (1 mg per kg) for 7 days. Tumors were collected and snap-frozen. For metabolite collection, tumor weights were first measured before processing samples for data normalization. Then tumors were homogenized in cold methanol, followed by sonication.

When sample collections were done, extractions were conducted following the biphasic extraction protocol. Basically, add cold chloroform to samples (in methanol) with a ratio of 2:1. Vortex samples for 1 min to homogenize. Add 1 fraction of water and vortex samples again. Glass vials were centrifuged at 3000 × $g$ for 10 min for phase separation. The aqueous phase was transferred to a new glass vial for metabolomics, and the remaining interphase was used for protein quantification. The aqueous phase was evaporated under nitrogen flow. Samples were resuspended in 50% acetonitrile, and the volume was scaled according to the protein amounts. In total, 15 μl was used for the lowest biomass, and all others were scaled accordingly. Standard mixes were prepared at 100 μM and run after the samples to allow for the identification of the targets.

Isotope labeling: a. U-$^{13}$C$_6$-glucose labeling. ICC cells were maintained in the indicated treatment for 24 h, at which point labeled media (glucose-free RPMI [Thermo Fisher Scientific, 11879020] supplemented with 10% dialyzed serum and 11 mM U-$^{13}$C$_6$-glucose [Sigma, 389374]) was replaced. For glucose-flux analysis, cells were cultured in the labeled media for 1 hour and 24 hours before harvest. b. U-$^{13}$C$_{16}$-palmitic acid labeling. 1 million ICC13-7 cells were seeded in a 6 cm dish in media with 10% lipid-depleted serum (Omega Scientific Inc., FB-50). The next day cells were treated with DMSO or 100 nM infigratinib for 24 hours, at which point labeled media supplemented with 100 μM Bovine Serum Albumin-U-$^{13}$C$_{16}$-palmitic acid (Cambridge Isotope Laboratories, Inc., CLM-409-PK) conjugate (1:2) and 1 mM L-carnitine (Sigma-Aldrich, C0283) was replaced. Cells were cultured in the labeled media for 8 h before harvest.

LC–MS/MS: All solvents were HPLC–MS grade from Sigma Aldrich. Metabolomics was performed by LC–MS using a Vanquish LC coupled to an ID-X MS (Thermofisher Scientific). 5 μL of sample or standard was injected on a ZIC-pHILIC peek-coated column (150 mm × 2.1 mm, 5 micron particles, maintained at 40 °C, Sigma-Aldrich). The LC program started at 93% Buffer B (Acetonitrile 97% in water) and 7% Buffer A (20 mM Ammonium Carbonate, 0.1% Ammonium hydroxide in water). Subsequent steps were 40% Buffer B/60% Buffer A in 19 min, 100% Buffer A in 9 min, 100% Buffer A for 5 min, then back to 93% Buffer B/7% Buffer A in 3 min and re-equilibration at 93% Buffer B/7% Buffer A for 9 min. A flow rate of 0.15 mL min$^{-1}$ was used, except other than during the initial 30 s (uniform ramping of flow rate from 0.05 to 0.15 mL min$^{-1}$). Data were acquired on the ID-X by switching polarities at 120,000 resolution, with an AGC target of 1e5, and a $m/z$ range of 65–1000. MS1 data were acquired in switching polarities for all samples and analyzed using Compound Discoverer 3.3 (ThermoFisher Scientific). A mix of standards for each target was run along with the samples and used as the unlabeled reference for the Compound Discoverer labeling workflow. Isotopomer distribution is found in the exchange column, represented as the percentage of 0 labeled carbon, 1 labeled carbon, 2 labeled carbon, etc. These values are corrected for natural abundance. Tracefinder was used to manually extract areas for compounds that could not be analyzed by the Compound Discoverer.

## Seahorse experiments

Cell Mito Stress Test: Oxygen consumption rates (OCR) and extracellular acidification rates (ECAR) were measured using XFe96 Extracellular Flux Analyzer (Agilent) according to the manufacturer's instructions with slight modifications. Briefly, ICC cells were cultured in 96-well Seahorse plates (Agilent) with indicated treatments. To analyze OCR and ECAR, the cells were washed with RPMI supplemented with 1%FBS, 2 mM glutamine, and 10 mM glucose 3 times. Measurements of OCR/ECAR were carried out at baseline and after injections of oligomycin (1.5 μM), FCCP/sodium Pyruvate (1 μM/1 mM), Rotenone (1.5 μM), and 2-Deoxyglucose (50 mM). All the measurements were normalized to the total protein amount measured by the BCA assay kit and analyzed with Wave software.

Mitochondrial Fuel Flex test: ICC13-7 and ICC21 cells were cultured in Seahorse XF RPMI in 96-well Seahorse plates (Agilent) with DMSO and Infigratinib for 24 hours. Measurements of OCR were conducted at baseline and after injections of 3 inhibitors (3 μM BPTES, 4 μM Etomoxir, and 2 μM UK5099) according to the manufacturer's protocol using XFe96 Extracellular Flux Analyzer. All the OCR measurements were normalized to the total protein amount measured by the BCA assay kit and analyzed with Wave software.

Dependency = [OCR (Baseline) − OCR (Target inhibitor)]/[OCR (Baseline) − OCR (All inhibitors)].

Capacity = 1 − [[OCR (Baseline) − OCR (Other 2 inhibitors)]/[OCR (Baseline) − OCR (All inhibitors)]].

Fatty acid oxidation measurement: ICC13-7 and ICC21 cells were cultured in Seahorse XF RPMI in 96-well Seahorse plates (Agilent) with DMSO and Infigratinib for 24 hours. Measurements of OCR were conducted at baseline and after injections of 4 μM Etomoxir.

Fatty acid oxidation rate = OCR (baseline) − OCR (Etomoxir).

### SLC2A1 dependency

SLC2A1 dependency was analyzed based on the Broad Institute Dependency Map (DepMap) (DepMap, Broad (2023). This DepMap release contains data from CRISPR knockout screens[57] from project Achilles, as well as genomic characterization data from the CCLE project. https://doi.org/10.6084/m9.figshare.22765112.v2). We compared SLC2A1 dependency between FGFR2-fusion+ ICC cell lines (ICC13-7, ICC10, and ICC10-6) and other 977 cancer cell lines. A dependency score of less than −0.5 indicates essentiality.

### Lipase activity assay

ICC13-7 cells were treated with DMSO and 100 nM infigratinib for 48 h. Lipase activity was measured with Lipase Activity Assay II (Sigma, MAK047) following the manufacturer's recommended protocol. Data normalized to cell number.

### Phosphoproteomics

ICC13-7 cells, treated with 75 nM futibatinib for 4 and 24 h in duplicates, were snap frozen. Then frozen ICC cell pellets were lysed, and the resulting proteins underwent reduction with DTT and alkylation with iodoacetamide. Precipitation was carried out following the MeOH/CHCl3 protocol, and the proteins were digested using LysC and trypsin. Phosphopeptide enrichment was performed as previously described[58]. Each sample, consisting of 2.5 mg of peptides, underwent phosphopeptide enrichment on TiO2 beads from GL Sciences, Japan. Subsequently, phosphopeptides were labeled with TMT-10plex reagents from Thermo Fisher Scientific, pooled, and fractionated into 24 fractions using basic pH reversed-phase chromatography, as outlined in previous studies[59].

The fractions were then dried, re-suspended in a solution of 5% ACN and 5% formic acid, and subjected to analysis in 3-hour runs via LC-M2/MS3 on an Orbitrap Fusion Lumos mass spectrometer. The simultaneous precursor selection (SPS) supported MS3 method was employed, as detailed in earlier publications[60–62]. Two MS2 spectra were acquired for each peptide using CID and HCD fragmentation methods. MS2 spectra were assigned using a SEQUEST-based, in-house-built proteomics analysis platform, allowing for phosphorylation of serine, threonine, and tyrosine residues as a variable modification[63]. The Ascore algorithm was utilized to assess the accurate assignment of phosphorylation within peptide sequences[64].

A target-decoy database search strategy was employed[65], and utilizing linear discriminant analysis and posterior error histogram sorting, peptide and protein assignments were filtered to achieve a false discovery rate (FDR) of less than 1%[66]. Peptides with sequences present in more than one protein sequence from the UniProt database (2014) were assigned to the protein with the highest number of matching peptides. For quantification, only MS3 with an average signal-to-noise value greater than 40 per reporter ion, along with an isolation specificity larger than 0.75[61], were considered. Protein TMT intensities were subject to a two-step normalization, first normalizing protein intensities over all acquired TMT channels for each protein based on the median average protein intensity calculated for all proteins. Correction for possible mixing errors was addressed by calculating a median of the normalized intensities from all protein intensities in each TMT channel, and the protein intensities were then normalized to the median value of these median intensities.

### ATGL ELISA measurement

ICC13-7 cells were treated with DMSO and 100 nM infigratinib for 48 h. ATGL protein levels were measured with a Human ATGL ELISA Kit (ab270890) following the manufacturer's recommended protocol. Data are normalized to cell number.

### NF-κB reporter assay

pNL3.2.NF-κB-RE[NlucP/NF-κB-RE/Hygro] Vector (Promega, N111A) was transiently transfected into ICC13-7 cells with TransIT®-LT1 Transfection Reagent (Mirus, MIR 2300) and selected with hygromycin to obtain a stably transfected cell line. The stable ICC13-7 cells were cultured in the 96 well plates and treated with DMSO and infigratinib for 24 h. Next, Nano-Glo® Luciferase Assay (Promega, N1110) was performed to measure the luciferase activity following the manufacturer's instruction. The data are normalized to cell numbers.

### siRNA transfection

Cells were seeded at 40% confluency in 6-well plates and allowed to attach overnight. The next day, siRNA transfections were performed with Lipofectamine RNAiMAX Transfection Reagent (Invitrogen, 13778075) according to the manufacturer's instructions. Fresh media were changed after 24 h of transfection. siRNAs were purchased from Dharmacon: siGENOME Human RELA siRNA-SMARTpool (#M-003533-02-0005), ON-TARGETplus Human MAP3K14 siRNA (j003580-14-0002, j003580-15-0002, j003580-17-0002), and siGENOME Non-targeting siRNA (#D-001210-01-05).

### Autophagic flux assay

ICC13-7 cells were engineered to stably express the LC3-GFP-mCherry construct. Then cells were subjected to DMSO or infigratinib treatment for 48 h. After treatment, cells were digested and resuspended in culture media. The fluorescence was measured by flow cytometry. The gating strategy was provided in Supplementary Fig. 10. The ratio of mCherry to GFP was calculated as autophagic flux. Data were analyzed using FlowJo (Tree Star, Ashland, OR) software.

### Crystal violet staining

ICC13-7 ($1 \times 10^5$ per well) and ICC21 ($6 \times 10^5$ per well) were seeded in six-well plates. The next day, the indicated treatments were performed. Fresh medium was changed every three days. 2 or 3 weeks after treatment, the medium was aspirated, and cells were washed with cold PBS, fixed with pre-cold methanol for 20 min at 4 °C and stained with 0.5% crystal violet in 25% methanol for 20 min at room temperature. Cells were then rinsed in tap water to remove extra color.

### Immunofluorescence

Ki67 staining was performed by iHisto Inc. (iHisto.io) using paraffin-embedded samples sectioned at 4 μm[23]. In brief, sections were deparaffinized and hydrated sequentially using xylene, 100% ethanol, 75% ethanol, and then PBS at room temperature. For antigen retrieval, slides were boiled in 10 mM sodium citrate for 10 min in a microwave oven and then cooled for 5 min at room temperature. Sections were rinsed in PBS three times, treated with 3% $H_2O_2$ for 15 min, and blocked

with 5% bovine serum albumin for 20 min. Sections were then incubated with Rabbit anti-Ki67 (1:200) and Mouse anti-panCK (1:200) overnight at 4 °C, followed by washing and incubation with CY3-conjugated goat anti-Rabbit (#A10520, Invitrogen, 1:2000) and FITC-conjugated Goat anti-Mouse (Thermo Fisher Scientific, Cat# A10520) for 50 min at room temperature. After incubation with the DAPI, whole slide scanning (20x) was performed (Panoramic midi scanner, 3D Histech). QuPath software was used to quantify Ki67-positive cells.

Mitochondrial morphology was assessed with MitoTracker dyes (Invitrogen, M-7514) in live cells according to the manufacturer's instructions and observed with a confocal microscope. For quantification, three morphological categories were defined: fragmented (cells with only fragmented mitochondria), elongated (cells with only elongated mitochondria), and intermediate (cells with both fragmented and elongated mitochondria). In total ~100 cells in each culture condition were counted.

LC3 staining was performed as previously described[67]. Cells were seeded in a 96-well plate. After 48 h treatment with 100 nM infigratinib, cells were fixed and permeabilized with ice-cold 100% methanol for 3 minutes. After PBS washing, coverslips were incubated in blocking buffer [PBS containing 5% (v/v) normal goat serum and 0.05% (w/v) saponin] for 1 h at RT and then incubated with LC3B antibody in the blocking buffer at 4 °C overnight. Cells were washed with PBS and then incubated with Alexa Fluor-conjugated secondary antibodies in a blocking buffer for 1 h at RT. Cells were rinsed with PBS and mounted with Vectashield for confocal microscopy.

Lipid droplet staining was performed with BODIPY Dye. Cells were seeded and grown in the 96-well plate. After treatments, cells were fixed with PBS containing 4%(v/v) paraformaldehyde (PFA) for 20 min, then stained with 2 μM 4,4-difluoro-1,3,5,7,8-pentamethyl-4-bora-3a,4a-diaza-s-indacene (BODIPY 493/503, Invitrogen, D3922) and 1 μg per ml Hoechst 33342 in PBS for 30 minutes at RT. After washing with PBS, cells are subjected to confocal microscopy.

Fluorescence images for LC3 and BODIPY staining were obtained with a Perkin Elmer Opera Phenix system using a 60 ×1.42 N.A. water objective. Images were captured with a Z stack of 1 mm per section for a total of two sections. After acquisition, the images were projected to form one image by maximum-intensity projection. The Harmony High-Content Imaging and Analysis Software was used for image analyses.

## Immunohistochemistry

Paraffin sections (4 μm) were baked at 37 °C for 1 h and washed in H$_2$O twice. Hydration was performed via the following steps: 5 min in xylene three times; 3, 3, 3, 1, 1, and 1 min in 100%, 100%, 90%, 75%, 50%, and 30% ethanol, respectively; and 3 min in H$_2$O. Antigen retrieval was achieved by boiling the sections in citrate solution (VECTOR, H-3300) for 20 min in a pressure cooker, followed by 20 minutes of cooling at room temperature. Sections were then washed with H$_2$O three times and treated with 3.5% H$_2$O$_2$ for 20 minutes and 10% normal goat serum (Life Technologies, 50062Z) for 1 h. The sections were incubated with primary antibodies Rabbit anti-Ki67 (1:200) overnight at 4 °C. Subsequently, the sections were immunohistochemically stained with anti-rabbit secondary antibodies (Biocare Medical, RHRP520H) for 1 h at room temperature. After incubation with the Betazoid DAB Chromogen Kit (Biocare Medical, BDB2004H). Slides were subject to H&E staining, and dehydration, and representative images were obtained by photography with a Nikon Eclipse Ti microscope. at 10X magnification. Quantitation was done by QuPath software.

## $^{18}$F-FDG-PET

The ICC13-7 xenografts (treated with vehicle or pemigatinib 1 mg per kg for 4 days) were fasted overnight before injection of 100–200 μCi $^{18}$F-FDG via the tail vein. During the uptake period (40–60 min), the mice were anesthetized under 1.5% isoflurane. PET imaging was recorded, followed by a CT scan using a micro PET/CT scanner (Inveon; Siemens, Germany). MicroQ Viewer software (Version 1.7.0.6; Siemens) was used to reconstruct PET data.

## Animal studies

Mice were maintained with standard protocols for laboratory animal care. Mice were housed in controlled environments to ensure their well-being and to maintain experimental consistency. The housing conditions adhered to established guidelines outlined in the Institutional Animal Care and Use Committee (IACUC) regulations (Protocol 2005N000148 and 2019N000116). Mice were monitored daily by a staff with dedicated expertise in research animal welfare.

The mice were housed in polycarbonate cages allowing five mice per cage in a dedicated animal facility at the Simches Research Building. Bedding material consisting of autoclaved wood shavings and was changed regularly, The environment had controlled temperature (20–24 °C) and humidity (40–60%) levels. Lighting conditions followed a standard 12-h diurnal cycle. Enrichment was provided in the form of nesting material. Mice were given ad libitum access to standard laboratory rodent chow (Prolab® IsoPro® RMH 3000, 5P75-RHI-W 23). Water was provided through an automatic watering system. All in vivo studies involved subcutaneous implant of ICC tissue or cells into 6- to 10-week-old NSG male and female mice (NOD.Cg-Prkdcscid Il2rgtm1Wjl/SzJ, 00557, The Jackson Laboratory). Tumor growth was monitored twice a week by measuring tumor size with a digital caliper and tumor volume was calculated twice a week with the formula (length × width$^2$)/2. The MGH IACUC regulations for maximum tumor size (<2 cm in greatest diameter) were strictly adhered to.

For ICC13-7/ICC21 xenografts, $3 \times 10^6 / 2 \times 10^6$ cells per mouse were injected. When tumors reached ~200 mm$^3$, mice were randomized and started on treatment. For vehicle, pemigatinib (1 mg per kg, daily), ONC212 (50 mg per kg, twice a week), or combination treatments, a fasting regimen was performed. Mice were withheld food for 24 h two times per week (with ad libitum access to water and hydrogel), separated by 2 or 3 days of ad libitum refeeding. ONC212 treatments were applied at the beginning of the refeeding phase. For vehicle, pemigatinib (1 mg per kg, daily), IACS-010759 (5 mg per kg, every other day), or combo treatments, a fasting regimen was performed that the mice were withheld food for 24 hours every other day (with ad libitum access to water and hydrogel), IACS-010759 treatments were applied during the fasting phase.

PDXs were used as described with slight changes[16]. When tumors reached ~300 mm$^3$, mice were randomized and started on treatment with vehicle and futibatinib (25 mg per kg, 14 days) or vehicle and infigratinib (15 mg per kg, 10 days) for MG69; vehicle and pemigatinib (1 mg per kg, 11 days) for MG212. All drugs were dosed by oral gavage.

Tumor samples were collected 4 h after the last treatment dose, and then portions were used snap-frozen for biochemical analysis or subjected to histology processing.

Formulations for each drug were as follows: infigratinib (PEG300/ D5W [2:1, v/v]), pemigatinib (5% DMAC in 0.5% methylcellulose aqueous solution), futibatinib (hydroxypropyl methyl cellulose solution), ONC212 (70% PBS, 10% DMSO, and 20% Kolliphor EL), and IACS-010759 (0.5% methylcellulose aqueous solution).

## Statistical analysis

GraphPad Prism 9 was used for statistical analysis and the generation of graphs. Data were presented as mean ± SD from biological replicates. Data between multiple groups (vehicle, single agents and both drugs) was analyzed by One-way ANOVA multiple comparisons. For two group comparisons, the Student's t-test (two-tailed) was performed. Values of $P < 0.05$ were considered significant. Statistical analyses and biological replicates from independent experiments are reported in the figure legends.

## Reporting summary

Further information on research design is available in the Nature Portfolio Reporting Summary linked to this article.

## Data availability

The RNA-Seq data reported in this study are deposited in the NCBI's Gene Expression Omnibus (GEO) database under accession code GSE241735. Publicly available RNA-Seq data used in this study are from the ArrayExpress database at EMBL-EBI under accession number E-MTAB-11324. The metabolomics data reported in this study are deposited to Metabolomics Workbench[68] (https://www.metabolomicsworkbench.org). The data can be accessed directly via Project PR001775 (https://doi.org/10.21228/M88M6V). All mass spectrometer raw files can be accessed through the MassIVE data repository (massive.ucsd.edu) under the accession number MSV000093916 (https://massive.ucsd.edu/ProteoSAFe/dataset.jsp?task=598187e239be46e3bfd7a26dc64a7a49). Source data are provided in this paper. All remaining data can be found in the Article, Supplementary, or Source data files. Source data are provided in this paper.

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

## Acknowledgements

We thank members of Bardeesy Lab for their technical support and valuable input. We thank the following core facilities: MGH next next-generation sequencing core, the Center for Comparative Medicine (CCM) at MGH, the Mouse Imaging Program (MIP) at the Center for Systems Biology at MGH, and Harvard Center for Mass Spectrometry. N.B. was supported by grants from the Samuel Waxman Cancer Research Foundation (jointly awarded to A.B.), the National Institute of Health (P50CA127003), TargetCancer Foundation, DOD (CA160216 and CA210849), and the Warren Alpert Foundation (W.S. El-Deiry, PI), and a V Foundation Translational Research Award. Y.Z. was supported by the Cholangiocarcinoma Foundation Mark R. Clements Memorial Research Fellowship. H.E. was supported by the Irving W. Janock Fellowship. Metabolomics Workbench is supported by NIH grant U2C-DK119886 and OT2-OD030544 grants. ONC212 was provided by Chimerix, Inc.

## Author contributions

Y.Z. and N.B. conceptualized the study, interpreted data, and wrote the manuscript, which was edited and approved by all authors. Y.Z., K.L., L.S., S.S. and Q.X. performed the experiments. H.E., E.B., A.B. and D.J. provided insights into experimental design and data interpretation. J.K., R.M. and W.H. performed a phosphoproteomics experiment and analysis.

## Competing interests

N.B. reports research grant support from Kinnate Biopharma, Tyra Biosciences, and Servier Laboratories. D.J. reports grants and personal fees from Novartis, Genentech, Syros, and Eisai, grants from Pfizer, Ribon Therapeutics, Infinity, InventisBio, Cyteir, and Arvinas, and personal fees from Relay Therapeutics, Vibliome, Mapkure, and PIC Therapeutics outside the submitted work. The remaining authors declare no competing interests.
