## [Peer Review File · Nature Communications]

Reviewers' Comments:

Reviewer #1:

Remarks to the Author:

The manuscript by Zhen Y. and colleagues aims to explore the transcriptional and metabolic consequences of FRFR inactivation in patient-derived models of FGFR2+ intrahepatic cholangiocarcinoma. The study integrates metabolic studies using tracing isotopes, multiomics strategies and cell signaling methodologies to identify a strong glycolytic phenotype in FGFR2-fusion+ ICC cells which is abrogated by FGFR2 inhibition. The authors further demonstrate that glycolysis reprogramming by FGFR2 requires NF- κ B pathway activation and that FGFR2 inhibition results in a compensatory metabolic switch to FAO, fueled by exogenously sourced FAs. The study is novel, interesting, robust, scholarly sound and well-written. There are few aspects that I strongly believe should be clarified. Please, refer to the specific comments below.

MAJOR CONCERNS:

- 1) While aerobic and anaerobic glycolysis are reduced, PPP remains unchanged. What fuels PPP in glucose scarcity conditions?
- 2) Could not find human FGFR2 in TRRUST, could you please include TRRUST version used and info on dataset used to generate Fig 3A panel?
- 3) Etomoxir, CPT-1 inhibitor, blocks FA flux to mitochondrial for FAO, fueled either by MUFA from lipid droplets or SFA (e.g., palmitate) via exogenous uptake or endogenous synthesis (DNL). Normally, SFA is preferentially channeled to mitochondria for FAO while MUFA are preferentially esterified into TG for storage in lipid droplets. The data presented in Fig. 4 seems to suggest that FA are not synthesized endogenously (Fig 4E) but rather are sourced from the exogenous pool and channeled to mitochondria via increased lipolysis (Fig. 4D). I believe that three major experiments should be performed to further clarify data in Fig 4 (FA for FAO): 1) Does lipolysis increase because of increased activity of ATGL? 2) ¹³C-Palmitate metabolic flux to really rule out endogenous vs exogenous origin; 3) provide assessment of lipid droplet breakdown by number/size of lipid droplets per cell or area, rather the fluorescence intensity which may not provide accurate quantification of lipid droplets.
- 4) A direct consequence of intensified FAO is ROS generation and potential mitochondrial damage. It seems that mitochondrial structure (based on MitoTracker data) did not show sign of mitochondrial damage. Does ROS generation increase? The unchanged PPP activation may function as defense mechanism for increased oxidative stress due to enhanced FAO.
- 5) The metabolic switch to FAO in FGFR2i treated cells is convincing. However, it is not clear what are the players promoting this switch and what promotes uptake of FAs or increased lipolysis to provide FAs for FAO.
- 6) Data on PDX, which is the most translational model of this study is limited. An in vivo or ex vivo assessment of glucose deprivation in PDXs would be more informative than a xenograft model using ICC13-7 cells that are not implanted in a liver orthotopic fashion.
- 7) The data that FGFR2 fusions drive NF- κ B activation are not compelling, and are based solely on transcriptomics. Rapid signal transduction events are seldom regulated by transcriptional processes. The authors should perform phospho-transcriptomic analysis to examine the mechanisms by which NF- κ B is activated. I would opine it is most likely activation of a cytokine receptors.
- 8) Finally, glucokinase regulated glucose metabolism in hepatocytes, whereas hexokinase regulates glucose metabolism in cholangiocytes. Do the observations imply that FGFR2-fusion cancers arise from cholangiocytes?

Reviewer #2:

Remarks to the Author:

This paper by Zhen and coll. entitled "FGFR inhibition blocks NF- κ B-dependent glucose metabolism and confers metabolic vulnerabilities in cholangiocarcinoma" reports high glycolytic activity in FGFR2-fusion+ cholangiocarcinoma (3 cell lines and 2 PDX were analyzed) that is dependent on NF- κ B, and decreased by FGFR inhibitors without affecting mitochondrial respiration which is supported by fatty acid oxidation and mitochondrial dynamics; this data led the authors to test combined treatments targeting both FGFR2 and mitochondrial metabolism in vitro and in vivo,

offering new perspectives in the clinic.

This manuscript is very well written and contains a significant amount of preclinical work supporting the interest of treatments combining chemotherapy with energy metabolism inhibition and diet intervention. However, it could be improved by addressing the following issues.

1. Overall, the rationale of the alternative use of different FGFR inhibitors is not clear. For example, in Figure 4, why Seahorse experiments were done using Infigratinib whereas gene expression experiments with Futibatinib?
2. Figure 2: in the isotope labelling metabolomics experiments, it is unclear why no difference in most metabolites is observed at the 24h time point.
3. Figure 2L: please explain what is the SLC2A1 gene effect score.
4. Page 8 line 193: please add that the RNA-seq data were used for this specific analysis.
5. Page 9 line 207: why not testing phospho-RELA also in cell lines?
6. Figure 4: the impact of FGFR inhibitors on other proteins than DRP1 involved in mitochondrial dynamics (in particular fusion: mitofusins 1 and 2 and OPA1) as well as on mitophagy should be tested.
7. Figure 5: decrease of tumor volume could result from cell death: apoptosis should be assessed by IHC (using activated-caspase 3 antibody for example).
8. Page 14 line 340: the sentence seems incomplete.

Reviewer #3:

Remarks to the Author:

In this interesting study, Zhen et al dissect the role of FGFR2-fusions in intrahepatic cholangiosarcoma (ICC). Using transcriptomics and metabolomics in FGFR2+ and FGFR2- ICC models in vitro and in vivo, authors unveil that FGFR inhibition leads to a strong glycolytic block, while maintaining OXPHOS intact. In addition, this glycolytic block seems to be subsequent to inhibition of the NFkB pathway, which is strongly inhibited by FGFR inhibition. Consistently, ICC cells are more dependent on fatty acid oxidation and autophagy. Finally, authors perform in vivo experiments using FGFR inhibitors in combination with ONC212 (a mitochondrial targeting agent) and found enhanced therapeutic effects with tumor regressions, both under control diet or using intermittent fasting (which reduces blood glucose levels). I think this is a nice study uncovering relevant biology in ICC. Still, I have some comments for authors to address (not listed in order of importance):

1. In Fig. 1H, I think it would be good to show GLUT1 levels (same as in neighboring panel Fig 1J). Also, I think it would be worthy to show PKM2 levels in both panels, as PKM is the rate limiting enzyme in glycolysis and it seems to be heavily affected too as seen in panel 1G.
2. Authors show transcriptomics data upon FGFRi in vitro and in vivo (Fig 1), but only show metabolomics data upon in vitro inhibition (Fig 2). Can they also show this?
3. Authors show FGFRi leads to reduced extracellular lactate (Fig 2F) while lactate intracellular levels seem to be not (or very minimally) affected (Fig 2D). Authors should discuss this.
4. In Fig 2I-K, authors should show labeling levels of all the different intermediates affected in Fig 1, not just the few selected in the present version.
5. Authors seem to favor the idea that FGFRi leads to reduced NFkB signaling, and this might be causing the glycolytic effect. At the very least, direct inhibition of NFkB leads to reduced glucose uptake and use (Fig 3 and Sup Fig 6). However, they don't directly prove this hierarchy. In addition, most FGFRi effects on glycolysis are assessed at 24h of treatment, whereas effects on NFkB are assessed as early as 1h post-treatment (Fig 3). Still, TCA incorporation of glucose-derived carbon seems to be affected also as early as 1h post-treatment (Fig 2J-K). Thus, it is not entirely clear to me whether NFkB is "only" upstream of the glycolytic phenotype, or whether the glycolytic phenotype might be also a direct effect of FGFRi and might in turn downregulate NFkB (which might then reinforce the phenotype by further inhibition of glycolysis). I think shorter-term experiments to try to dissociate the first effects on glycolysis and NFkB signaling are needed.

In line with this, would a constitutively active form of NFkB rescue the effects of FGFRi?

6. Authors should show LC3 activation levels via WB too in Sup Fig 7, not only the fluorescence measurements in Fig 4K-L.

7. Authors seem to believe they get enhanced/synergistic effects in Fig 5 with intermittent fasting. However, this is difficult to gauge right now because the experiment with regular diet is buried in Sup Fig 8, and the main figure 5 only shows intermittent fasting. Related to this, it is worth noting that authors see regressions with the combo treatment under both types of diets. In addition, the rate of growth of vehicle-treated tumors seems to be vastly different between diets (~140% in regular diet vs 40% in intermittent fasting). Thus, it is difficult to interpret the effect of the combo under both diets. An approximation to this might be investigating what is the fold difference in volume in combo vs vehicle mice in both diets. But right now with the data presented I'm not sure if intermittent fasting drives better responses to the combo treatment, or it "only" has an intrinsic effect on tumor growth irrespective of treatment.

Reviewer #1

The manuscript by Zhen Y. and colleagues aims to explore the transcriptional and metabolic consequences of FRFR inactivation in patient-derived models of FGFR2+ intrahepatic cholangiocarcinoma. The study integrates metabolic studies using tracing isotopes, multiomics strategies and cell signaling methodologies to identify a strong glycolytic phenotype in FGFR2-fusion+ ICC cells which is abrogated by FGFR2 inhibition. The authors further demonstrate that glycolysis reprogramming by FGFR2 requires NF- κ B pathway activation and that FGFR2 inhibition results in a compensatory metabolic switch to FAO, fueled by exogenously sourced FAs. The study is novel, interesting, robust, scholarly sound and well-written. There are few aspects that I strongly believe should be clarified. Please, refer to the specific comments below.

We thank the reviewer for generously taking the time to evaluate our manuscript and for their thoughtful comments, which we address below.

MAJOR CONCERNS:

1) While aerobic and anaerobic glycolysis are reduced, PPP remains unchanged. What fuels PPP in glucose scarcity conditions?

While we did not observe a change in labeling of PPP metabolites in the ¹³C-glucose isotopic tracing experiment at the time points analyzed, we did find a drop in steady state levels after 24 hours FGFRi treatment (**Figure 2d** of the manuscript). Since labeling into the PPP is a fast process, it is possible that examination of shorter labeling times would have detected a change. We clarified this point in the Results section. We agree that elucidating the pathways supporting the PPP in these cells is of interest, although we would need to conduct dynamic flux analysis to fully elucidate these networks. Since we do not see marked changes in ROS, it appears that REDOX changes are not central to the proliferative arrest resulting from FGFRi treatment under the conditions studied.

2) Could not find human FGFR2 in TRRUST, could you please include TRRUST version used and info on dataset used to generate Fig 3A panel?

We apologize if we were not clear in the presentation of these data or if we have misunderstood the reviewer's question. We used TRRUST to predict which transcription factors govern the FGFR2-regulated transcriptome in FGFR2-fusion+ ICC cells, based on differentially expressed genes after FGFR inhibitor treatment. The output is a list of transcription factors (in our case, showing enrichment of multiple NF κ B subunits). We have updated the manuscript to clarify this. For this analysis, we used Version 2 of TRRUST to interrogate the following RNAseq datasets: FGFRi sensitive models (ICC13-7, MG69, MG212 +/- FGFRi treatment) first introduced in Figure 1, and partially sensitive and resistant models (ICC21, ICC10-6 +/- dual FGFRi+EGFRi treatment) first introduced in Supplementary Figure 2. We have now clarified these information in the revised manuscript.

Please note that the TRRUST database has two query formats, one to predict enriched transcription factors over a large dataset and the other to predict which transcription factors are candidate regulators of a single input gene, based on its regulatory sequence. The current version of the TRRUST database provides predicted regulators of FGFR1, but not FGFR2. Since we were not using the database for the purpose of identifying transcriptional regulators of FGFR2, we do not mention this aspect.

If we have not understood the reviewer correctly or if further information is requested, we would be happy to address any further concerns as to this point.

3) Etomoxir, CPT-1 inhibitor, blocks FA flux to mitochondrial for FAO, fueled either by MUFA from lipid droplets or SFA (e.g., palmitate) via exogenous uptake or endogenous synthesis (DNL). Normally, SFA is preferentially

channeled to mitochondria for FAO while MUFA are preferentially esterified into TG for storage in lipid droplets. The data presented in Fig. 4 seems to suggest that FA are not synthesized endogenously (Fig 4E) but rather are sourced from the exogenous pool and channeled to mitochondria via increased lipolysis (Fig. 4D). I believe that three major experiments should be performed to further clarify data in Fig 4 (FA for FAO): 1) Does lipolysis increase because of increased activity of ATGL? 2) 13C-Palmitate metabolic flux to really rule out endogenous vs exogenous origin; 3) provide assessment of lipid droplet breakdown by number/size of lipid droplets per cell or area, rather the fluorescence intensity which may not provide accurate quantification of lipid droplets.

i) Does lipolysis increase because of increased activity of ATGL?

We thank the reviewer for these important points. First, we examined the impact of FGFRi treatment on the mRNA and protein expression of the key lipases, ATGL (encoded by PNPLA2), HSL (encoded by LIPE) and MGL (encoded by MGLL). We found that FGFRi treatment did not significantly affect the expression of ATGL/PNPLA2 (the rate-limiting lipase) or of HSL/LIPE, and only modestly affected MGLL levels in ICC13-7 cells (**Reviewer Figure 1a**). Moreover, PNPLA2 protein levels were unchanged as shown by ELISA (**Reviewer Figure 1b**, incorporated into **manuscript Supplementary Figure 7c**)

Reviewer Figure 1. Inhibition of FGFR signaling has a limited effect on lipase mRNA expression levels in FGFR2-fusion+ ICC cells.

a, Relative mRNA levels of lipase genes in ICC13-7 cells treated with DMSO or 75 nM futibatinib for 4 hours and 24 hours. **b**, PNPLA2 protein levels were measured by ELISA (Abcam) in ICC13-7 cells treated with 100 nM infigratinib for 48 hours. Data represent means \pm SD from independent experiments. One-way ANOVA multiple comparisons were performed within each group for (a) and two-tailed Student's t-test was performed for (b).

Subsequently, to examine signaling changes, we performed global quantitative mass spectrometry-based phosphoproteomics on FGFR2-fusion+ ICC cells treated with vehicle or FGFRi. We analyzed cells treated for 4hrs to identify the more acute signaling events and at 24hr to gain insight into adaptive changes. These studies detected 18,103 phosphoforms (phosphopeptides with a unique phosphorylation pattern representing 19,136 phosphorylation sites) from 3,954 proteins (**Reviewer Figure 2a**, incorporated into **manuscript Figure 3f**). We first conducted pathway analysis of the differential tyrosine phosphorylated proteins as an initial validation of the experiment. As expected, decreased tyrosine phosphorylation of FGFR signaling proteins was the most prominent pathway change at both 4 and 24 hours FGFRi treatment (**Reviewer Figure 2b-c**).

Reviewer Figure 2. Global quantitative mass spectrometry-based phosphoproteomics on FGFR2-fusion+ ICC cells treated with vehicle or FGFRi. a, Schematic of phosphoproteomics analysis. b-c, Pathway enrichment analysis (BioPlanet database) shows that Tyrosine phosphorylation of established FGFR targets is strongly downregulated by 75 nM futibatinib treatment of ICC13-7 cells for 4 hours (b) or 24 hours (c). The top-ranked pathways are shown.

Our prior studies indicated that MAPK pathway activation was the central output of oncogenic signaling by FGFR2 fusions in ICC, with downstream activation of mTOR/p70S6K signaling¹. To further validate the analysis and to globally determine changes in the activity profile of the kinome, we queried the site-specific changes of phosphoserine- and phosphothreonine-carrying peptides (pS/T) against the new Kinase Library platform. This kinome-wide dataset predicts the kinases capable of phosphorylating each phosphorylation site across the human Serine/Threonine phosphoproteome² (the equivalent platform for tyrosine kinases is under development by the Cantley lab). Enrichment analysis thus allows the identification of the serine/threonine kinases whose activity is altered by FGFRi treatment (**Reviewer Figure 3a**, incorporated into **manuscript Supplementary Figure 5g**). We confirmed that the activity of the ERK/RSK/p70S6K axis was strongly downregulated at both 4 hours and 24 hours of treatment (**Reviewer Figure 3b-c**, incorporated into **manuscript Supplementary Figure 5h-i**).

Reviewer Figure 3. Global quantitative mass spectrometry-based phosphoproteomics on FGFR2-fusion+ ICC cells treated with vehicle or FGFRi. a, Schematic of phosphoproteomics coupled with prediction of Ser/Thr kinase activity. b-c, Kinase enrichment analysis of pS/T-carrying peptides using the Kinase Library platform predicts the kinases whose activity is most downregulated in ICC13-7 cells in response to 75 nM futibatinib treatment for 4 hours (b) or 24 hours (c). The top ranked pathways are shown.

These data reinforce ERK and its targets as a primary output of FGFR2 fusions, extending our prior signaling studies^{3,4}.

We predicted that the analysis of the early and later time points of FGFR inhibition (4h and 24h) would provide information about both acute signaling changes proximal to FGFR2/ERK activity as well as adaptive signaling changes reflecting gradual alterations in the cell state. Accordingly, we next interrogated lipase enzymes for phosphorylation changes at these time points. Notably, we found that FGFRi treatment resulted in significantly increased phosphorylation of PNPLA2 (Ser-404) and LIPE (Ser-855 and Ser-895) emerging specifically at 24 hours of treatment (**Reviewer Figure 4a-b**, incorporated into **manuscript Figure 4h-i**). Phosphorylation of the mouse PNPLA2 paralog at Ser-406 (corresponding to human PNPLA2 Ser-404) by the energy-sensing kinase, AMPK, has been reported to stimulate lipolysis^{5,6}. Similar regulation has been suggested for human ATGL⁷. While the functional significance of the other phosphorylation changes has not been evaluated, it is notable that LIPE (Ser-855) is also a predicted AMPK site (**Reviewer Figure 4c-d**, incorporated into **manuscript Supplementary Figure 7d-e**). No changes were observed in phosphorylation of PNPLA2 (S428) and LIPE (S951), which are not predicted AMPK substrates (**Reviewer Figure 4b**).

Reviewer Figure 4. FGFRi treatment results in a gradual increase in the phosphorylation of lipase proteins. a, Volcano plot indicating phosphoproteome changes in ICC13-7 cells treated with 75 nM futibatinib for 24 hours. Phosphorylation sites that are significantly upregulated (>1.5 -fold change and $p < 0.05$) are shown in blue. Red dots highlight significantly upregulated phosphosites on lipases (gene name and site position are indicated). b, Bar graphs quantifying the phosphorylation changes of lipase proteins at 4- and 24-hour treatment of 75 nM futibatinib. c-d, Predicted kinases for the indicated phosphorylation events. $\text{Log}_2(\text{Score}) > 0$ is considered favorable kinases for a given site; $\text{Log}_2(\text{Score}) < 0$ indicates unfavorable kinases. Top 5 ranked-favorable kinases are listed. AMPKs were labeled bold. Data represent means \pm SD from independent experiments. One-way ANOVA multiple comparisons were performed within each group for (b).

Importantly, we demonstrated that FGFRi treatment stimulated lipase activity (**Reviewer Figure 5**, incorporated into **manuscript Figure 4g**). Collectively, these data are consistent with a role for increased ATGL-mediated lipolysis in the metabolic adaptation to FGFRi treatment. Further extensive studies that we feel to be beyond the scope of the current manuscript would be required to fully establish the mechanisms of lipase regulation. Nevertheless, the phosphoproteome enrichment data are in line with the observed phenotypes and we include these data in the manuscript for the reviewers' consideration.

Reviewer Figure 5. FGFR inhibition leads to increased lipase activity. Lipase activity was tested by Lipase Activity Assay Kit II (Sigma) in ICC13-7 cells treated with DMSO or 100 nM infigratinib for 48 hours. Data represent means \pm SD from independent experiments. A two-tailed Student's t-test was performed.

ii) 13C-Palmitate metabolic flux to really rule out endogenous vs exogenous origin

We have performed U- $^{13}\text{C}_{16}$ -palmitate tracing in FGFR2+ ICC cells +/- FGFRi treatment. ICC13-7 cells were treated with DMSO or infigratinib for 24 hours in lipid-depleted media, and then switched to ^{13}C -palmitate labeling media for another 8 hours (with DMSO or infigratinib). Cells were then harvested for metabolomics (**Reviewer Figure 6a**, incorporated into **manuscript Supplementary Figure 7f**). FGFR inhibition did not affect the generation of palmitate-derived TCA cycle metabolites (**Reviewer Figure 6b**, incorporated into **manuscript Supplementary Figure 7g**). These data suggest that endogenous lipids rather than exogenous lipids may be the primary source of fatty acids supplying FAO upon FGFRi treatment. However, since there are additional sources of lipids in serum, including lipoproteins, we cannot categorically rule out a role for exogenous uptake and report our conclusions accordingly.

Reviewer Figure 6. FGFR inhibition does not affect TCA cycle labelling from exogenous fatty acids. a, Schematic of workflow for the $^{13}\text{C}_{16}$ -palmitate tracing experiment. b, Percentage of ^{13}C enrichment for different TCA cycle metabolites (M+2) after $^{13}\text{C}_{16}$ -palmitate labeling for 8 hours in ICC13-7 cells. Cells were pre-treated with 100 nM infigratinib or DMSO for 24 hours prior to labeling and maintained under these conditions during the labeling step. Data represent means \pm SD from independent experiments, and two-tailed Student's t-test was performed.

iii) assessment of lipid droplet breakdown by number/size of lipid droplets per cell or area

We now present data from BODIPY staining and confocal microscopy which shows that FGFRi treatment results in the progressive and striking decrease in lipid droplet number from 24 to 48 hours, but mild change on size (**Reviewer Figure 7**, incorporated into **manuscript Figure 4f**).

Reviewer Figure 7. Lipid droplets are decreased by FGFRi treatment. a-b, BODIPY staining of ICC13-7 cells treated with DMSO or 100 nM infigratinib for 24 hours and 48 hours. Number (a) and size (b) of lipid droplets were calculated with confocal microscopy. Data represent means \pm SD from independent experiments. One-way ANOVA multiple comparisons were performed.

Thus, the data indicate that FGFRi treated increases FAO through the coordinated increase in liberation of endogenous stored lipids.

4) A direct consequence of intensified FAO is ROS generation and potential mitochondrial damage. It seems that mitochondrial structure (based on MitoTracker data) did not show sign of mitochondrial damage. Does ROS generation increase? The unchanged PPP activation may function as defense mechanism for increased oxidative stress due to enhanced FAO.

We assessed mitochondrial damage by testing for changes in mitochondrial membrane potential using Tetramethylrhodamine, methyl ester (TMRM) staining. No significant changes were observed upon FGFRi treatment (**Reviewer Figure 8a**). We also did not observe changes in studies measuring mitochondrial ROS using the MitoSOX reagent (**Reviewer Figure 8b**).

To test the potential protective role of sustained REDOX control in response to FGFRi, we co-treated cells with 6-Aminonicotinamide (6-AN) to inhibit G6PD and thereby limit PPP activity. Assessment of cell viability after 7 days showed that there was an additive effect of combined FGFRi and 1 μ M 6-AN under the conditions assayed (**Reviewer Figure 8c**). Overall, the data are consistent with a contribution of sustained PPP activity in mitigating any ROS increases and preserving cell viability in the context of FGFRi treatment, although redox control does not appear to be a primary read-out of the FGFR pathway under the conditions studied.

Reviewer Figure 8. Inhibition of FGFR signaling has a limited effect on mitochondria ROS state. a-b, ICC13-7 cells were treated with 100 nM infigratinib for 24h, then staining with TMRM (a) and MitoSOX (b). c, ICC13-7 cells were treated with DMSO, 100 nM infigratinib, 1 mM 6-AN or combination for 7 days. MTT (3-(4,5-dimethylthiazol-2-yl)-2,5-diphenyltetrazolium bromide) assay was performed to measure cell viability. Data represent means \pm SD from independent experiments. Two-tailed Student's t-test (a-b) and one-way ANOVA multiple comparisons (c) were performed.

5) The metabolic switch to FAO in FGFR2i treated cells is convincing. However, it is not clear what are the players promoting this switch and what promotes uptake of FAs or increased lipolysis to provide FAs for FAO.

As noted above (response to point #3), we observed that FGFRi treatment results in the gradual induction of activating phosphorylation of ATGL (p-S404) as well increased phosphorylation of other less characterized sites on HSL (S855, S895). These changes involve known (ATGL- S404) or predicted AMPK sites (and predicted sites for the closely related AMPK family kinases: SIK and BRSK) (**Reviewer Figure 4**, above). Collectively, our findings provide the foundation for further studies that dissect the mechanisms underlying metabolic adaptations to FGFRi treatment.

In addition, consistent with the palmitate labeling indicating that FGFRi treatment did not affect fatty acid uptake (**Reviewer Figure 6**, above), FGFRi treatment broadly did not affect or slightly decreased mRNA expression levels of fatty acid uptake factors (including transport proteins [FATPs], binding proteins [FABPs] and receptors). The exceptions were a modest upregulation of the transporter SLC27A3 and CPT1B (both expressed at very low levels [CPM <1]) and ACSS1 (Acetyl-CoA synthetase 1, contributing to mitochondrial uptake) also exhibited a small increase in expression (**Reviewer Figure 9**, incorporated into **Supplementary Figure 7h**).

Reviewer Figure 9. FGFRi treatment does not cause significant induction of fatty acid uptake factors.

RNAseq data of the indicated factors in ICC13-7 cells treated with fufibatrinib (75 nM, 24 hours) or DMSO. Data were normalized to the vehicle condition and presented as Log₂ transformation.

6) Data on PDX, which is the most translational model of this study is limited. An in vivo or ex vivo assessment of glucose deprivation in PDXs would be more informative than a xenograft model using ICC13-7 cells that are not implanted in a liver orthotopic fashion.

We thank the reviewer for this comment. We would first like to clarify that the ICC13-7 model is a patient-derived cell line that our lab has established (and which we maintain as a low-passage model). We agree with the reviewer on the importance of expanding our in vivo preclinical studies and now include data from a second patient-derived model (ICC21, harboring the CBX5 fusion). This model was subjected to the same protocol

incorporating intermittent fasting in the treatment regimen (**Reviewer Figure 10a**, incorporated into **Supplementary Figure 8k**). Whereas FGFRi treatment (pemigatinib, 1 mg per kg) slowed growth and ONC212 had no effect, the combination potentiates the tumor growth arrest (**Reviewer Figure 10b**, incorporated into **Supplementary Figure 8l**). Thus, we observe comparable efficacy of combined mitochondrial targeting and FGFR inhibition in xenografts generated from two different patient-derived models of FGFR2-fusion+ ICC.

Reviewer Figure 10. Targeting mitochondria metabolism coupled with intermittent fasting increases the effectiveness of FGFR inhibitor treatment against FGFR2 fusion+ ICC xenografts. a, Schematic of intermittent fasting and treatment regimen. b, Mice harboring ICC21 xenografts (starting tumor volume ~200 mm³) were treated with vehicle (n=5), 1 mg per kg pemigatinib (n=7), 50 mg per kg ONC212 (n=5), or the combination of both drugs (n=6) for 14 days under the intermittent fasting regimen. Tumor volume changes are shown. Data represent means ± SD. One-way ANOVA multiple comparisons were performed.

Our patient-derived cell lines were generated as part of a larger effort to create a biliary tract cancer cell line atlas comprising cell lines of the common genotypes of this cancer. A key question was whether these *in vitro* models retain the molecular circuitry of the tumor types from which they were derived. To address this, we first compared the transcriptomes of a matched set of 22 cell lines with their associated PDX model (established directly from patient tumor samples). We found that 16/22 (~73%) of the cell lines were the nearest neighbors with the respective PDXs, including each of our FGFR2 fusion+ models, such as ICC13-7 and ICC21, shown by two-dimensional uniform manifold approximation (UMAP) (**Reviewer Figure 11**). Thus, despite adaptation to 2D culture and passaging *in vitro*, the BTC cell lines broadly maintained transcriptional programs characteristic of the *in vivo* tumors.

Reviewer Figure 11. The transcriptional profiles of bile duct cancer cell lines recapitulate those of associated PDX models. Two-dimensional representation of RNA sequencing data of cell lines and matched PDX tissues (after filtering source effects) using uniform manifold approximation and projection (UMAP) demonstrates high concordance between PDX tissues (triangles) and corresponding cell lines (circles). The FGFR2+ ICC models used in the present study are highlighted.

We agree that the PDX experiment is certainly of value; however, for a 4-arm experiment, it would take more than 9 months to conduct this study, given the need to significantly expand the tumor material and reimplant and grow up to be suitable for treatment. Therefore, we hope that our inclusion of a second patient-derived cell line-based xenograft (CDX) model will be sufficient in providing increased robustness and rigor to the study.

7) The data that FGFR2 fusions drive NF- κ B activation are not compelling, and are based solely on transcriptomics. Rapid signal transduction events are seldom regulated by transcriptional processes. The authors should perform phospho-transcriptomic analysis to examine the mechanisms by which NF- κ B is activated. I would opine it is most likely activation of a cytokine receptors.

We thank the reviewer for this comment and apologize for not being clear in how we stated our conclusions. Indeed, in line with the reviewer's comment, our conclusion is that FGFR2 fusions regulate NF- κ B activity primarily via more direct control of NF- κ B signaling. We observe rapid changes in the activity state of NF- κ B pathway signaling proteins upon FGFR inhibition. These changes include a decrease in p-IKK α / β (S176/180) and an associated increase in I κ B α (an IKK target that inhibits NF- κ B), observed within 1 hour of treatment (**Manuscript Figure 3d** and **Reviewer Figure 12a**, included below for the reviewer's convenience). We likewise show that inhibition of the downstream effectors of FGFR2 fusions, SHP2 and MEK, causes similar inhibitor effects on the NF- κ B pathway signaling (decreasing p-IKK α / β [S176/180]), and we confirmed that trametinib suppressed NF- κ B target gene expression, implicating the SHP2/MEK axis a critical mediator of FGFR2-driven NF- κ B activation (**Manuscript Supplementary Figure 5e-f** and **Reviewer Figure 12b-c**, included below for the reviewer's convenience).

Reviewer Figure 12. FGFR2/SHP2/MEK inhibition leads to a rapid decrease in NF- κ B pathway signaling. a-c, Immunoblot analysis of NF- κ B signaling proteins in ICC13-7 cells treated with 100 nM infigratinib (a) or either 100 nM GDC-1971 or 100 nM trametinib (b) for the indicated time points. c, Relative mRNA levels of established NF- κ B targets in ICC13-7 cells treated with 100 nM trametinib or DMSO for 4 hours. Data represent means \pm SD from independent experiments. Two-tailed Student's t-test

Our Phosphoproteomics data also highlighted the interplay between FGFR2 and NF- κ B signaling. Notably, unbiased gene set enrichment analysis (EnrichR platform) of pS/T site-specific changes revealed that phosphorylation of NF- κ B components was significantly reduced by 4 hours FGFRi treatment (**Reviewer Figure 13a-b**, incorporated into **manuscript Figure 3g-h**). Phosphorylation decreases were observed for the IKK inhibitory protein, TNIP1 (at Ser-619 and Ser-627, both predicted ERK sites), the receptor TNFRSF21 (at Ser-525, also a very strongly predicted ERK sites, and Ser-521 and Ser-541, which are additional potential ERK sites), and for NF- κ B2 (at Ser-161, a predicted S6K site) (**Reviewer Figure 13c-f**, incorporated into **manuscript Supplementary Figure 5j-m**). Phosphorylation at the non-ERK/S6K sites, TNFRSF21 (S565) and NF- κ B2 (S222), did not change (**Reviewer Figure 13c**, incorporated into **manuscript Supplementary Figure 5n**).

Reviewer Figure 13. Phosphorylation of NF- κ B components is significantly reduced by FGFRi treatment. a, Unbiased pathway enrichment analysis (Hallmark database) shows that Ser/Thr phosphorylation of NF- κ B components is downregulated by 75 nM futibatinib treatment of ICC13-7 cells for 4 hours. The top 4 pathways are shown. b, Volcano plot indicating Ser/Thr phosphoproteome changes in ICC13-7 cells treated with 75 nM futibatinib for 4 hours. Red dots highlight significantly downregulated phosphosites on NF- κ B components (gene name and site position are indicated; phosphosites that are predicted ERK or S6K targets are labeled bold). c-f, Predicted kinases for the indicated phosphorylation events. g, Normalized phosphorylation level of NF- κ B components with predicted ERK/S6K and non-ERK/S6K sites. Data represent means \pm SD from independent experiments. One-way ANOVA multiple comparisons were performed.

With respect to the functional role of the cytokine receptors, we considered TNFRSF21, which is robustly expressed across models and shows marked changes in ERK-site phosphorylation (but no change in the non-ERK site S565) in response to FGFRi treatment (**Reviewer Figure 13g**, above). Given these data suggesting the potential regulation of TNFRSF21 upon FGFRi treatment and the high relative level of expression of this factor in FGFR2-fusion+ ICC cells, we addressed its potential functional role in NF- κ B regulation. To this end, we used siRNA to knock down TNFRSF21 in FGFR2+ ICC cells. We observed that TNFRSF21 silencing resulted in decreased activating phosphorylation of IKK α / β (i.e. phospho-Ser[176/180]) and accumulation of the inhibitor protein, I κ B α , associated with a reduction in NF- κ B target genes (**Reviewer Figure 14 a-b**). We present these data as a reviewer-only figure as they further link FGFR2-ERK signaling to the NF- κ B pathway, although it would require further in-depth study to understand mechanistically.

Reviewer Figure 14. TNFRSF21 stimulates NF- κ B signaling. a-b Immunoblot analysis of NF- κ B pathway proteins (a) and relative mRNA expression of the indicated NF- κ B targets (b) in ICC13-7 cells transfected with siRNA against TNFRSF21 or control. Data represent means \pm SD from independent experiments. Two-tailed Student's t-test was

While we conclude that FGFR2 regulates NF- κ B by rapid, direct signaling processes, as suggested by the reviewer, we do note that there are decreases mRNA expression of components of the NF- κ B pathway (**Manuscript Supplementary Figure 5o** and **Reviewer Figure 15**, included below for the reviewer's convenience). These decreases are likewise delayed, presumably reinforcing the NF- κ B pathway is overall inactivation upon FGFRi treatment.

Reviewer Figure 15. Impact of FGFRi treatment on NF- κ B pathway components. RNA-seq data upon treatment of ICC13-7 cells with DMSO or 75 nM futibatinib for 4 hours.

Our data show that FGFRi treatment (and downstream SHP2/MEK suppression) leads to rapid inactivation of the NF- κ B signaling pathway, as reflected by modulation of upstream of key signaling components (loss of activating IKK phosphorylation, increased I κ B α stability), associated with decreased phosphorylation of additional pathway components. Further studies are needed to fully establish the critical direct mechanisms.

8) Finally, glucokinase regulated glucose metabolism in hepatocytes, whereas hexokinase regulates glucose metabolism in cholangiocytes. Do the observations imply that FGFR2-fusion cancers arise from cholangiocytes?

We thank the reviewer for this interesting question. We find that GCK expression is very low or absent in all HCC (Liver) and cholangiocarcinoma (Bile Duct) cell lines (median $\log_2(\text{TPM}+1) < 0.1$ for Bile Duct, and even lower for Liver) whereas HK2 expression is robust across cell lines from both cancer types (median $\log_2(\text{TPM}+1) > 3.5$ for both) (**Reviewer Figure 16a-b**). Thus, HK2 rather than GCK is expressed in cancer cells regardless of cell of origin.

Reviewer Figure 16. HK2 exhibits markedly higher mRNA expression than GCK across cell lines from both HCC and bile duct cancer. a-b The RNA expression of GCK and HK2 in the liver/HCC (a) and bile duct/cholangiocarcinoma cell lines (b) from Cancer Cell Line Encyclopedia (CCLE) data.

Again, we would like to thank the reviewer for taking the time to conduct this thorough review and we hope we have adequately addresses their questions and concerns.

Reviewer #2

This paper by Zhen and coll. entitled “FGFR inhibition blocks NF- κ B-dependent glucose metabolism and confers metabolic vulnerabilities in cholangiocarcinoma” reports high glycolytic activity in FGFR2-fusion+ cholangiocarcinoma (3 cell lines and 2 PDX were analyzed) that is dependent on NF- κ B, and decreased by FGFR inhibitors without affecting mitochondrial respiration which is supported by fatty acid oxidation and mitochondrial dynamics; this data led the authors to test combined treatments targeting both FGFR2 and mitochondrial metabolism in vitro and in vivo, offering new perspectives in the clinic.

This manuscript is very well written and contains a significant amount of preclinical work supporting the interest of treatments combining chemotherapy with energy metabolism inhibition and diet intervention. However, it could be improved by addressing the following issues.

We thank the reviewer for their supportive comments and their valuable critiques, which we address in our point-by-point response below.

1. Overall, the rational of the alternative use of different FGFR inhibitors is not clear. For example, in Figure 4, why Seahorse experiments were done using Infigratinib whereas gene expression experiments with Futibatinib?

We thank the reviewer for this question. During the course of the work three different inhibitors were approved for FGFR2+ ICC (pemigatinib, infigratinib, and futibatinib). They are all based on the same chemotype, sharing a bulky dimethoxyphenyl ring group, which is critical to the activity of this group of inhibitors (at the bottom of the structures depicted in **Reviewer Figure 17**). As shown in our previous studies⁴, they produce comparable effects *in vitro* and *in vivo* against FGFR2+ ICC (the primary difference in the activity of these molecules is their ability to overcome FGFR2 kinase domain mutations, which are not relevant in the present study). Given their structural similarity and high selectivity for FGFR1-4, we have used these molecules interchangeably across our experiments. We have been careful to indicate which compound was used for each study.

			Name	Infigratinib	Futibatinib	Pemigatinib
IC50 (nM)	3~8.2	0.3~1.5	5~12.5

Reviewer Figure 17. FGFR inhibitors share the same chemotype with similar IC50 (half maximal inhibitory concentration) against FGFR2-driven ICC cells. The chemical structure of the FGFR inhibitors used in the manuscript and their respective IC50 against FGFR2-fusion driven ICC cells (CCLP; FGFR2-PHGDH fusion model¹). Futibatinib has an additional covalent binding mode.

2. Figure 2: in the isotope labelling metabolomics experiments, it is unclear why no difference in most metabolites is observed at the 24h time point.

We thank the reviewer for this question. This reflects the speed of various metabolic processes. To evaluate the more rapid glucose utilization in glycolysis, TCA cycle, and PPP, we conducted labeling for 1 hour. The data showed that these pathways become saturated by 24 hours as expected. The later time point allows analysis of nucleotide synthesis which takes longer to reach a steady state⁸.

3. Figure 2L: please explain what is the SLC2A1 gene effect score.

The Gene Effect Score is a metric devised for the Broad Institute DepMap platform (<https://depmap.org/portal/>) to measure the dependence of a cell line on a given gene. Lower values indicate dependency, with score below -0.5 considered significant dependencies. We have added the explanation in the revised text.

4. Page 8 line 193: please add that the RNA-seq data were used for this specific analysis.

We apologize for not being clear. We interrogated the TRRUST dataset with the same RNA-seq data as in Figure 1, and have now included this clarification in the revised text.

5. Page 9 line 207: why not testing phospho-RELA also in cell lines?

We find that phosphorylated IKK α/β , total I κ B α , and NF- κ B target gene expression are consistent read-outs upon FGFRi treatment across our models, whereas p-RELA (S536) changes vary, with marked changes in PDXs *in vivo* and no changes *in vitro* in the models tested (**Reviewer Figure 18**). The primary read-out for a blocking of the NF- κ B pathway is downregulation of p-IKK and upregulation of I κ B α . By contrast, RELA (S536)

phosphorylation is more complex since it is both a marker of activation and degradation (showing oscillating levels despite upstream pathway activation) and since there are additional kinases for this site, casein kinase $\gamma 1$ (CK1 $\gamma 1$) and TBK1⁹.

Reviewer Figure 18. FGFR inhibitor treatment suppresses NF- κ B signaling in FGFR2-fusion+ ICC models. Immunoblot analysis of NF- κ B pathway proteins in FGFR2-fusion+ ICC models. PDX MG69 was treated with vehicle or infigratinib (15 mg per kg) for 10 days. PDX MG212 was treated with vehicle or pemigatinib (1 mg per kg) for 11 days. ICC13-7 cells were treated with DMSO or 100 nM infigratinib for the indicated times.

6. Figure 4: the impact of FGFR inhibitors on other proteins than DRP1 involved in mitochondrial dynamics (in particular fusion: mitofusins 1 and 2 and OPA1) as well as on mitophagy should be tested.

We have checked the expression of each of these factors. We do not find marked changes at 4 hours and 24 hours infigratinib treatment in ICC13-7 cell line (**Reviewer Figure 19**, incorporated into **manuscript Supplementary Figure 7i**).

Reviewer Figure 19. Inhibition of FGFR signaling has limited effect on the expression of Mito-fusion proteins. Immunoblot analysis of the indicated mito-fusion proteins in ICC13-7 cells treated at 4 hours and 24 hours with infigratinib (100 nM).

We also examined for mitochondrial damage by testing for changes in mitochondrial membrane potential using Tetramethylrhodamine, methyl ester (TMRM) staining and for mitochondrial ROS using the MitoSOX reagent. No significant changes were observed upon FGFRi treatment (**Reviewer Figure 20**).

Reviewer Figure 20. Inhibition of FGFR signaling has a limited effect on mitochondria membrane potential and mitochondrial ROS state. a-b, ICC13-7 cells were treated with 100 nM infigratinib for 24h, then stained with TMRM (a) and MitoSOX (b). Data represent means \pm SD from independent experiments. Two-tailed Student's t-test was performed.

As for mitophagy, we tested the colocalization of LC3 and mitochondria (stained by TOM20). Consistent to no mitochondria damage, we did not observe changes in mitophagy upon FGFR inhibition for 24 hours (**Reviewer Figure 21**).

Reviewer Figure 21. Inhibition of FGFR signaling does not have strong effects on mitophagy. ICC13-7 cells were treated with DMSO or ifingratinib for 24 hours and stained with LC3 and TOM20 antibodies. The co-localization of LC3 and TOM20 was calculated by confocal microscopy. Data represent means \pm SD from independent experiments. Two-tailed Student's t-test was performed.

7. Figure 5: decrease of tumor volume could result from cell death: apoptosis should be assessed by IHC (using activated-caspase 3 antibody for example).

We have assessed cleaved caspase-3 via immunohistochemistry. We do observe an evident trend toward induction of caspase cleavage but the data do not reach statistical significance (**Reviewer Figure 22**). We provide the results here as reviewer-only data, but can incorporate into the manuscript if advised by the reviewers and editors.

Reviewer Figure 22. Cell death is not significantly induced by combination FGFRi and ONC212 treatment in the context of fasting in FGFR2 fusion+ ICC. Mice harboring ICC13-7 xenografts with a starting tumor volume \sim 200 mm³ were treated with vehicle (n=5), pemigatinib 1 mg per kg (n=6), ONC212 50 mg per kg (n=4), or the combination of both drugs (n=8) for 15 days under the intermittent fasting regimen. Quantification of IHC staining for cleaved-caspase 3 was shown. Data represent means \pm SD. One-way ANOVA multiple comparisons were performed.

8. Page 14 line 340: the sentence seems incomplete.

We thank the reviewer and have corrected the sentence to read:

“The FDA approval of FGFR tyrosine kinase inhibitors (TKIs) has been a significant advance in the treatment of FGFR2+ ICC.”

Again, we are grateful to the reviewer for generously taking their time to evaluate our manuscript and we hope that we have satisfied their concerns.

Reviewer #3

In this interesting study, Zhen et al dissect the role of FGFR2-fusions in intrahepatic cholangiosarcoma (ICC).

Using transcriptomics and metabolomics in FGFR2+ and FGFR2- ICC models in vitro and in vivo, authors unveil that FGFR inhibition leads to a strong glycolytic block, while maintaining OXPHOS intact. In addition, this glycolytic block seems to be subsequent to inhibition of the NFkB pathway, which is strongly inhibited by FGFR inhibition. Consistently, ICC cells are more dependent on fatty acid oxidation and autophagy. Finally, authors perform in vivo experiments using FGFR inhibitors in combination with ONC212 (a mitochondrial targeting agent) and found enhanced therapeutic effects with tumor regressions, both under control diet or using intermittent fasting (which reduces blood glucose levels). I think this is a nice study uncovering relevant biology in ICC. Still, I have some comments for authors to address (not listed in order of importance):

We are grateful to the review for their thorough review of our manuscript.

1. In Fig. 1H, I think it would be good to show GLUT1 levels (same as in neighboring panel Fig 1J). Also, I think it would be worthy to show PKM2 levels in both panels, as PKM is the rate limiting enzyme in glycolysis and it seems to be heavily affected too as seen in panel 1G.

We have included these data in the revised manuscript. We find that GLUT1 is reduced in the MG69 model, and to a lesser extent in ICC13-7 cells but is not changed in the MG212 PDX. PKM2 was modestly decreased in MG69 but not in the other models (**Reviewer Figure 23**, incorporated into **manuscript Figure 1h-j**).

Reviewer Figure 23. FGFR inhibitor treatment suppresses glycolytic protein expression in FGFR2-fusion+ ICC models. a-c, Immunoblot analysis of glycolytic proteins in MG212 (a), MG69 (b), and ICC13-7 (c). PDX MG69 was treated with vehicle or infigratinib (15 mg per kg) for 10 days. PDX MG212 was treated with vehicle or pemigatinib (1 mg per kg) for 11 days. ICC13-7 cells were treated with DMSO or 100 nM infigratinib for 24 hours.

2. Authors show transcriptomics data upon FGFRi in vitro and in vivo (Fig 1), but only show metabolomics data upon in vitro inhibition (Fig 2). Can they also show this?

We thank the reviewer for this important suggestion. We have treated xenografts generated with the patient-derived ICC13-7 model with vehicle or pemigatinib (1 mg per kg daily for 7 days) and measured steady-state metabolites. The data show a broad decrease in metabolites from central carbon metabolism, with strong enrichment for components of glycolysis (**Reviewer Figure 24a**). Notable changes included drops in 6-PG, PEP, GlcNA-1-P, and FBP, as well as dTTP among others (**Reviewer Figure 24b-d**). Together with our prior data showing decrease glucose uptake (¹⁸FDG-PET imaging) upon FGFRi treatment, these findings support a major role of the FGFR2-fusion in the metabolic reprogramming of ICC tumors. We have now included **Reviewer Figure 24a-b** in the manuscript as **Figure 5c-d**.

Reviewer Figure 24. Glucose metabolism is suppressed upon FGFR inhibition *in vivo*.

Metabolite levels were determined by LC-MS of ICC13-7 xenografts from mice treated with pemetigatinib (1 mg per kg) or vehicle for 7 days. a, Analysis of enriched metabolic pathways by the MetaboAnalyst platform. b, Volcano plot indicating the metabolite changes. Metabolites in glucose metabolism showing significant changes are represented as red dots and are identified by name. c-d, Heatmap depicting metabolite changes from different pathways. Data were normalized to the vehicle condition and presented as log₂ transformation.

3. Authors show FGFRi leads to reduced extracellular lactate (Fig 2F) while lactate intracellular levels seem to be not (or very minimally) affected (Fig 2D). Authors should discuss this.

We thank the reviewer for this comment. Since the lactate transporters (primarily MCT4) are highly efficient, lactate is typically rapidly secreted. This pattern of changed glycolysis but no change in intracellular lactate is observed upon KRAS inactivation in pancreatic cancer cells for example¹⁰.

4. In Fig 2I-K, authors should show labeling levels of all the different intermediates affected in Fig 1, not just the few selected in the present version.

We have now included the different intermediates in the revised manuscript (**Supplementary Figure 3g-k**).

5. Authors seem to favor the idea that FGFRi leads to reduced NFκB signaling, and this might be causing the glycolytic effect. At the very least, direct inhibition of NFκB leads to reduced glucose uptake and use (Fig 3 and Sup Fig 6). However, they don't directly prove this hierarchy.

In addition, most FGFRi effects on glycolysis are assessed at 24h of treatment, whereas effects on NFκB are assessed as early as 1h post-treatment (Fig 3). Still, TCA incorporation of glucose-derived carbon seems to be affected also as early as 1h post-treatment (Fig 2J-K). Thus, it is not entirely clear to me whether NFκB is "only" upstream of the glycolytic phenotype, or whether the glycolytic phenotype might be also a direct effect of FGFRi and might in turn downregulate NFκB (which might then reinforce the phenotype by further inhibition of

glycolysis). I think shorter-term experiments to try to dissociate the first effects on glycolysis and NFκB signaling are needed. In line with this, would a constitutively active form of NFκB rescue the effects of FGFRi?

We apologize for any lack of clarity. In fact, our studies of TCA incorporation of glucose-derived carbon in **Figure 2k** were performed 24 hours post-treatment of FGFRi. The designation of the 1 hour time point refers to the length of time after addition of labelled glucose. As requested by the reviewer, we now include steady state metabolomics analysis from a shorter time point (4 hours of FGFRi treatment). We find that whereas there are robust and widespread decreases in glycolytic and TCA cycle intermediates observed at 24 hours treatment (e.g. **Reviewer Figure 25a** and **manuscript Figure 2d**), very limited changes in metabolite levels are seen at 4 hours (**Reviewer Figure 25b-e**).

Reviewer Figure 25. Metabolomics analysis of FGFRi treated ICC cells. a, Normalized glycolytic intermediates levels were determined by LC-MS of ICC13-7 cells treated with 100 nM infigratinib or DMSO for 24 hours. b-d, Normalized metabolite levels from different pathways were determined by LC-MS of ICC13-7 cells treated with 100 nM infigratinib or DMSO for 4 hours. Data represent means \pm SD from independent experiments. Two-tailed Student's t-test was performed.

Regarding the kinetics of NF-κB regulation, we find that NF-κB signaling changes are rapid, with a notable drop in pIKK levels evident at 1-hour FGFRi treatment, whereas HK2 mRNA and protein expression changes are more delayed (**Reviewer Figure 26**). These data are consistent with NF-κB being upstream of decreases in HK2 and glycolytic activity.

Reviewer Figure 26. Short term FGFR inhibition has no effect on glycolytic protein expression. Immunoblot analysis of indicated proteins in ICC13-7 cells treated with 100 nM infigratinib or DMSO for 1 hour.

We have also mapped the changes in the phosphoproteome upon FGFRi treatment, which showed significant enrichment of phosphorylation of NF- κ B pathway components at early time points, with multiple of the phosphosites predicted to be targets MAPK/mTOR signaling, the major output of FGFR2-fusions, as discussed in detail in our response to Reviewer #1. In brief, we conducted global quantitative mass spectrometry-based phosphoproteomics on FGFR2-fusion+ ICC cells treated with vehicle or FGFRi for 4 hours to identify the more acute signaling events and at 24 hours to gain insight into adaptive changes. These analyses highlighted the strong downregulation of ERK and mTOR/S6-Kinase signaling upon FGFRi treatment (based on the use of the new Kinase Library Platform² to predict the kinases responsible for Serine/Threonine phosphorylation across the phosphoproteome (see **Reviewer Figures 2 and 3**, above).

Notably, unbiased gene set enrichment analysis (EnrichR platform) of pS/T site-specific changes revealed that phosphorylation of NF- κ B components was significantly reduced by 4 hours FGFRi treatment (**Reviewer Figure 27a**, incorporated into **manuscript Figure 3g**). Phosphorylation decreases were observed for the IKK inhibitory protein, TNIP1 (at Ser-619 and Ser-627, both predicted ERK sites based on Kinase Library analysis), the receptor TNFRSF21 (at Ser-525, a very strongly predicted ERK sites, and Ser-521 and Ser-541, which are also potential ERK sites), and for NF- κ B2 (at Ser-161, a predicted S6K site) (**Reviewer Figure 27b-f**, incorporated into **manuscript Supplementary Figure 5j-m**). Phosphorylation at the non-MAPK/S6K sites, TNFRSF21 (S565) and NF- κ B2 (S161), did not change (**Reviewer Figure 27g**, incorporated into **manuscript Supplementary Figure 5n**).

Collectively the data are in keeping with FGFR2-MEK-ERK regulating the NF κ B pathway via direct signaling mechanisms.

Reviewer Figure 27. Phosphorylation of NF- κ B components is significantly reduced by FGFRi treatment. a, Unbiased pathway enrichment analysis (Hallmark database) shows that Ser/Thr phosphorylation of NF- κ B components is downregulated by 75 nM futibatinib treatment of ICC13-7 cells for 4 hours. The top 4 pathways are shown. b, Volcano plot indicating Ser/Thr phosphoproteome changes in ICC13-7 cells treated with 75 nM futibatinib for 4 hours. Red dots highlight significantly downregulated phosphosites on NF- κ B components (gene name and site position are indicated; phosphosites that are predicted ERK or S6K targets are labeled bold). c-f, Predicted kinases for the indicated phosphorylation events. g, Normalized phosphorylation level of NF- κ B components with predicted ERK/S6K and non-ERK/S6K sites. Data represent means \pm SD from independent experiments. One-way ANOVA multiple comparisons were performed.

To further examine the hierarchy as requested by the reviewer, we have used 2-deoxyglucose to inhibit glycolysis in FGFR2-driven ICC cells and then tested for effects on NF- κ B. Cells were treated with 50 mM 2DG for 3 hrs and analyzed by western blot for NF- κ B signaling. We did not observe any changes in p-IKK (**Reviewer Figure 28**).

Reviewer Figure 28. Glycolysis inhibition does not activate NF-κB pathway. Immunoblot analysis of showing that IKK phosphorylation is unchanged in ICC13-7 cells treated with 50 mM 2-DG for 3 hours. AMPK activation (p-T172) is shown as a marker for 2-DG induced energy stress.

With respect to the role of NF-κB as the sole regulator of the phenotype, we found that genetic and pharmacological inhibition of NF-κB impaired glycolysis in FGFR2+ ICC cells (**manuscript Figure 3m-q, and Supplementary Figure 6a-d**). These data indicated the NF-κB is necessary for this phenotype. We have also conducted a series of rescue experiments, including forced expression of NIK (MAP3K14) and constitutively of active IKKα and IKKβ, and treatment with TNFα. We found that the ectopic expression studies (lentiviral transduction of MAP3K14, activated IKKα [S176E] and IKKβ [S180E]) alone or combined) failed to boost NF-κB signaling in FGFR2+ ICC cells, and did not rescue glycolytic gene expression. Subsequently, we tested media supplementation with 100 ng per ml TNFα, which effectively activated the pathway, inducing canonical inflammatory genes (**Reviewer Figure 29a-b**), but it not rescue glycolytic gene expression (**Reviewer Figure 29c**). While optimization of experimental conditions, such as testing different time points, might have produced a different result, these preliminary data indicate the NF-κB is insufficient to drive the glycolytic program in this context. This may reflect the widespread changes in the phosphoproteome and other biological shifts resulting from FGFRi treatment that could affect the regulation of specific gene targets. We have included further discussion of these points in the revised manuscript (Discussion).

Reviewer Figure 29. TNFα supplement does not rescue the decrease HK2 level resulting from FGFR inhibition. a-c, ICC13-7 cells were treated with 100 nM infigratinib for 24 hours, adding 100 ng per ml TNFα 4 hours before harvesting. Quantitative real-time PCR was performed to measure mRNA level of indicated genes. Data represent means ± SD from independent experiments. Two-tailed Student's t-test was performed.

6. Authors should show LC3 activation levels via WB too in Sup Fig 7, not only the fluorescence measurements in Fig 4K-L.

We have now performed the western blots which corroborate these findings on LC3 activation. We find that FGFRi treatment results in an increase in LC3B-II (assessed at 24 hrs treatment), whereas LC3B-I levels are at very low levels in these cells (**Reviewer Figure 30**, incorporated into the **manuscript Supplementary Figure 7I**).

Reviewer Figure 30. Inhibition of FGFR signaling upregulates autophagy. Immunoblot analysis of LC3B protein in ICC13-7 cells treated with 100 nM infigratinib or DMSO for 24 hours.

7. Authors seem to believe they get enhanced/synergistic effects in Fig 5 with intermittent fasting. However, this is difficult to gauge right now because the experiment with regular diet is buried in Sup Fig 8, and the main figure 5 only shows intermittent fasting. Related to this, it is worth noting that authors see regressions with the combo treatment under both types of diets. In addition, the rate of growth of vehicle-treated tumors seems to be vastly different between diets (~140% in regular diet vs 40% in intermittent fasting). Thus, it is difficult to interpret the effect of the combo under both diets. An approximation to this might be investigating what is the fold difference in volume in combo vs vehicle mice in both diets. But right now with the data presented I'm not sure if intermittent fasting drives better responses to the combo treatment, or it "only" has an intrinsic effect on tumor growth irrespective of treatment.

We have now included this analysis as requested and we find that there is a significant improvement in combination treatment when administered in the context of the fasting regimen (**Reviewer Figure 31**, incorporated into the **manuscript Supplementary Figure 8h**).

Reviewer Figure 31. Response to FGFRi + ONC212 treatment is potentiated by intermittent fasting. Normalized volume changes of combination treatment (pemigatinib plus ONC212) treatment relative to vehicle treatment of ICC13-7 xenografts in fed (n=8) and fasting condition (n=8) are shown. Data represent means \pm SD. Two-tailed Student's t-test was performed.

We are very grateful for the reviewer's valuable input. We hope that we have satisfied their concerns.

References

- 1 Wu, Q. *et al.* Landscape of Clinical Resistance Mechanisms to FGFR Inhibitors in FGFR2-Altered Cholangiocarcinoma. *Clin Cancer Res* **30**, 198-208 (2024). <https://doi.org:10.1158/1078-0432.Ccr-23-1317>
- 2 Johnson, J. L. *et al.* An atlas of substrate specificities for the human serine/threonine kinome. *Nature* **613**, 759-766 (2023). <https://doi.org:10.1038/s41586-022-05575-3>
- 3 Goyal, L. *et al.* TAS-120 Overcomes Resistance to ATP-Competitive FGFR Inhibitors in Patients with FGFR2 Fusion-Positive Intrahepatic Cholangiocarcinoma. *Cancer Discov* **9**, 1064-1079 (2019). <https://doi.org:10.1158/2159-8290.Cd-19-0182>
- 4 Wu, Q. *et al.* EGFR Inhibition Potentiates FGFR Inhibitor Therapy and Overcomes Resistance in FGFR2 Fusion-Positive Cholangiocarcinoma. *Cancer Discov* **12**, 1378-1395 (2022). <https://doi.org:10.1158/2159-8290.Cd-21-1168>
- 5 Kim, S. J. *et al.* AMPK Phosphorylates Desnutrin/ATGL and Hormone-Sensitive Lipase To Regulate Lipolysis and Fatty Acid Oxidation within Adipose Tissue. *Mol Cell Biol* **36**, 1961-1976 (2016). <https://doi.org:10.1128/MCB.00244-16>
- 6 Ahmadian, M. *et al.* Desnutrin/ATGL is regulated by AMPK and is required for a brown adipose phenotype. *Cell Metab* **13**, 739-748 (2011). <https://doi.org:10.1016/j.cmet.2011.05.002>
- 7 Awad, D. *et al.* Adipose triglyceride lipase is a therapeutic target in advanced prostate cancer that promotes metabolic plasticity. *Cancer Res* (2023). <https://doi.org:10.1158/0008-5472.Can-23-0555>
- 8 Antoniewicz, M. R. A guide to (13)C metabolic flux analysis for the cancer biologist. *Exp Mol Med* **50**, 1-13 (2018). <https://doi.org:10.1038/s12276-018-0060-y>
- 9 Christian, F., Smith, E. L. & Carmody, R. J. The Regulation of NF- κ B Subunits by Phosphorylation. *Cells* **5** (2016). <https://doi.org:10.3390/cells5010012>
- 10 Ying, H. *et al.* Oncogenic Kras maintains pancreatic tumors through regulation of anabolic glucose metabolism. *Cell* **149**, 656-670 (2012). <https://doi.org:10.1016/j.cell.2012.01.058>

Reviewers' Comments:

Reviewer #1:

Remarks to the Author:

The authors have comprehensively addressed all my prior comments and concerns.

Reviewer #2:

Remarks to the Author:

The authors have satisfactorily addressed the issues I had raised.

I suggest them to incorporate (Reviewer Figure 22) into the manuscript since the trend toward induction of caspase cleavage is evident and interesting (in Supplementary Figure 8).

Reviewer #3:

Remarks to the Author:

I would like to congratulate the authors for their very nice study

Reviewer #4:

Remarks to the Author:

The authors have addressed many of the reviewers' comments from the first round of review, and the overall result is a good manuscript that should be interesting to a fairly broad audience in the signaling / metabolism / and cancer fields. With that said, there are a few points that should be addressed for transparency and to strengthen the manuscript overall.

1. Only selected subsets of data are available in the supplemental tables. The authors need to provide a list of all phosphopeptides identified, along with quality scores, quantification under different conditions and replicates, etc. This should be one of the supplemental tables. Similar data needs to be made available for the metabolomics studies.
2. Many, if not most, of the figures in the rebuttal are for reviewer only. These data need to be made available to the readers of the manuscript, many of whom will have the same questions or concerns that were raised during review. These additional 'reviewer-only' data need to be included in either the main figures or as supplemental data/supplemental figures.
3. The inconsistent use of inhibitors is not well explained or justified.
4. Western blots need to be quantified and presented with statistical analysis of the data.

Reviewer #1 (Remarks to the Author):

The authors have comprehensively addressed all my prior comments and concerns.

Reviewer #2 (Remarks to the Author):

The authors have satisfactorily addressed the issues I had raised. I suggest them to incorporate (Reviewer Figure 22) into the manuscript since the trend toward induction of caspase cleavage is evident and interesting (in Supplementary Figure 8).

We have followed the reviewer's suggestion and incorporated the data into manuscript Supplementary Figure 8k.

Reviewer #3 (Remarks to the Author):

I would like to congratulate the authors for their very nice study

Reviewer #4 (Remarks to the Author):

The authors have addressed many of the reviewers' comments from the first round of review, and the overall result is a good manuscript that should be interesting to a fairly broad audience in the signaling / metabolism / and cancer fields. With that said, there are a few points that should be addressed for transparency and to strengthen the manuscript overall.

1. Only selected subsets of data are available in the supplemental tables. The authors need to provide a list of all phosphopeptides identified, along with quality scores, quantification under different conditions and replicates, etc. This should be one of the supplemental tables. Similar data needs to be made available for the metabolomics studies.

As requested in the author checklist, we have deposited the datasets into public data repositories (Metabolomics: DOI: <http://dx.doi.org/10.21228/M88M6V>; Phosphoproteomics: <https://massive.ucsd.edu/ProteoSAFe/dataset.jsp?task=598187e239be46e3bfd7a26dc64a7a49>).

2. Many, if not most, of the figures in the rebuttal are for reviewer only. These data need to be made available to the readers of the manuscript, many of whom will have the same questions or concerns that were raised during review. These additional 'reviewer-only' data need to be included in either the main figures or as supplemental data/supplemental figures.

The great majority of experimental data in the rebuttal is indeed presented in the manuscript (19 of the Rev. Figures), and the rest addresses minor/peripheral points, with much of the data giving a negative result. We prefer not to include extra data emerging from preliminary observations, which would require additional study for rigor and interpretation.

We understand that documents from the peer review process will be available publicly, associated with the article.

3. The inconsistent use of inhibitors is not well explained or justified.

We have added text to further clarify this in the Methods (with a guide to this text in the Results). As we mentioned, we used the three FGFR inhibitors approved by the FDA to treated FGFR2+ ICC. All are from a common chemotype (based on Infigratinib) and are highly selective and potent against FGFR1-3 at the doses used in our studies.

4. Western blots need to be quantified and presented with statistical analysis of the data.

We have noted in the legends the number of times that each has been repeated.